# Viral sequence determines HLA-E-restricted T cell recognition of hepatitis B surface antigen

Gavuthami Murugesan[1,2], Rachel L. Paterson [1,2], Rakesh Kulkarni [1], Veronica Ilkow [1], Richard J. Suckling [1], Mary M. Connolly[1], Vijaykumar Karuppiah [1], Robert Pengelly [1], Archana Jadhav [1], Jose Donoso [1], Tiaan Heunis [1], Wilawan Bunjobpol[1], Gwilym Philips[1], Kafayat Ololade[1], Daniel Kay[1], Anshuk Sarkar[1], Claire Barber [1], Ritu Raj[1], Carole Perot[1], Tressan Grant [1], Agatha Treveil [1], Andrew Walker [1], Marcin Dembek [1], Dawn Gibbs-Howe[1], Miriam Hock [1], Ricardo J. Carreira [1], Kate E. Atkin [1], Lucy Dorrell[1], Andrew Knox [1], Sarah Leonard [1], Mariolina Salio [1] & Luis F. Godinho [1] ✉

The non-polymorphic HLA-E molecule offers opportunities for new universal immunotherapeutic approaches to chronic infectious diseases. Chronic Hepatitis B virus (HBV) infection is driven in part by T cell dysfunction due to elevated levels of the HBV envelope (Env) protein hepatitis B surface antigen (HBsAg). Here we report the characterization of three genotypic variants of an HLA-E-binding HBsAg peptide, $Env_{371-379}$, identified through bioinformatic predictions and verified by biochemical and cellular assays. Using a soluble affinity-enhanced T cell receptor (TCR) (a09b08)-anti-CD3 bispecific molecule to probe HLA-E presentation of the $Env_{371-379}$ peptides, we demonstrate that only the most stable $Env_{371-379}$ variant, L6I, elicits functional responses to a09b08-anti-CD3-redirected polyclonal T cells co-cultured with targets expressing endogenous HBsAg. Furthermore, HLA-E-$Env_{371-379}$ L6I-specific CD8[+] T cells are detectable in HBV-naïve donors and people with chronic HBV after in vitro priming. In conclusion, we provide evidence for HLA-E-mediated HBV Env peptide presentation, and highlight the effect of viral mutations on the stability and targetability of pHLA-E molecules.

Approximately 2 billion people worldwide are infected with the hepatotropic Hepatitis B virus (HBV), with nearly 300 million being chronically infected. The annual mortality associated with chronic HBV infection is around 820,000 people, primarily from cirrhosis and hepatocellular carcinoma (HCC)[1,2]. HCC is the sixth most common cancer and the fourth leading cause of cancer-related death globally, with almost 50% of cases associated with chronic HBV infection[3–5]. Although a highly effective vaccine is widely available and has led to a decline in acute and chronic HBV infections (CHB) worldwide, ~1.5 million new infections occur annually, and up to 15% of vaccinated individuals do not acquire protection against HBV infection[2,6].

CHB is defined by the detection of circulating envelope protein hepatitis B surface antigen (HBsAg) or HBV DNA for more than 6 months. The major impediments to achieving a cure for HBV are the persistence of covalently closed circular DNA (cccDNA) located in the hepatocyte nucleus, which serves as a transcriptional template

---

[1]Immunocore Ltd, 92 Park Drive, Abingdon, Oxfordshire OX14 4RY, UK. [2]These authors contributed equally: Gavuthami Murugesan, Rachel L. Paterson.
✉e-mail: luis.godinho@immunocore.com

for various HBV RNAs, and the integration of HBV DNA into the host genome, which is a source of secreted HBsAg. Currently approved therapies for CHB are (1) nucleos(t)ide analogs (NAs), which suppress HBV replication but do not eliminate the viral reservoir and thus are a lifelong treatment; (2) pegylated interferon alpha-2a, which can drive cccDNA degradation leading to a cure in ~10% patients following a finite course of therapy but with limited use due to severe adverse effects. These treatments reduce the risk of cirrhosis, liver failure, and HCC, but rarely lead to sustained loss of HBsAg and undetectable HBV DNA in serum after completion of treatment, which are pre-requisites for a functional cure[7–9]. Thus, there is an urgent need for novel therapeutic approaches that can achieve the elimination of viral reservoirs and sustained off-treatment responses following a finite course of therapy with a more tolerable adverse event profile[7,8].

There is a strong rationale for including immunotherapeutic agents in a combination strategy with direct-acting antivirals in order to achieve a functional cure for HBV[10]. The majority (90–95%) of adults with acute HBV infections[3,11] mount broad, highly functional HBV-specific T cell responses and subsequently control infection. In contrast, CHB infection is associated with functional exhaustion of HBV-specific T cell responses[12–14]. Various strategies to reinvigorate HBV-reactive T cells are being pursued, such as Toll-like receptor agonists, immune checkpoint receptor inhibitors, and therapeutic vaccines[15]. However, HBV-specific T cells may be too exhausted to respond to such approaches.

The Immune mobilizing monoclonal T cell receptors Against Viruses (ImmTAV) platform offers a unique therapeutic approach that bypasses exhausted T cells by redirecting functional T cells to HBV-positive hepatocytes, irrespective of their specificity. ImmTAV molecules are bispecific T cell-engaging fusion proteins comprising an affinity-enhanced TCR specific for a viral peptide, fused to an anti-CD3 single-chain antibody variable fragment (scFv). The TCR binds to peptide–HLA complexes on the target cell surface with high affinity, while the anti-CD3 domain recruits and activates T cells regardless of specificity, eliciting the release of cytokines and cytotoxicity against the target cell[16–18]. This technology thus avoids the need for expansion of rare antigen-specific T cells. The bispecific TCR-anti-CD3 mechanism of action has been recently validated by the approval of tebenta-fusp, which targets a peptide derived from the melanocyte antigen, gp100, and confers a significant survival benefit in HLA-A*02:01-positive patients with metastatic uveal melanoma[19]. We previously developed an ImmTAV molecule that targets an HLA-A*02:01-restricted HBV Env-derived peptide and showed that it can selectively eliminate HBsAg-positive hepatocytes in vitro[17]. A clinical ImmTAV candidate is currently being tested in a phase I clinical trial in HLA-A*02:01 patients with hepatitis B e Antigen (HBeAg)-negative non-cirrhotic CHB[20,21]. However, the HLA-A*02:01 allele frequency is less than 30% in high CHB burden regions. Therefore, to maximize potential population coverage, we undertook a search for suitable peptide targets presented by the highly conserved HLA class Ib molecule HLA-E, of which only two equally expressed dominant alleles have been described, HLA-E*01:01 and HLA-E*01:03[22–24]. HLA-E typically binds nonameric peptides derived from HLA class Ia signal sequences (known as VL9), leading to its stabilization on the cell surface for innate recognition by NK cells[25–28]. However, HLA-E may occasionally present pathogen-derived peptides, thereby triggering an adaptive immune response against the pathogen[29–35].

In this work, we perform a systematic search for HLA-E binding peptides within the HBV proteome, using bioinformatics, HLA-E peptide-binding assays, and refolded HLA-E complexes. Having identified suitable candidate peptides, we isolate TCRs against three genotypic variants of HBV Env$_{371-379}$ and generate an ImmTAV molecule with picomolar affinities, which redirects polyclonal T cells to lyse HBV-associated hepatocellular carcinoma (HCC) cells and in vitro HBV-transfected cells expressing one of the genotypic variants (L6I$_{371-379}$). We also provide evidence for the naturally occurring presentation of this epitope through in vitro expansion of HLA-E-restricted CD8+ T cells specific for the HBV envelope L6I$_{371-379}$ peptide in HBV-naïve donors and people with chronic HBV. These studies demonstrate the potential strengths and limitations of immune-mediated targeting of HLA-E-presented viral peptides to achieve a functional cure for CHB.

## Results

### HLA-E binds the HBV Env$_{371-379}$ peptide and its genotype variants

A bioinformatic approach was used to identify peptides derived from the HBV Env proteins (large, middle, and small hepatitis B surface proteins) that could bind HLA-E. We searched for peptides using netMHCpan4.0 affinity prediction and sequence conservation across the five main HBV genotypes (A, B, C, D, E). Overall, 70 potential HBV Env HLA-E binding peptides (9-mer and 10-mer, Supplementary Data 1) from these HBV genotypes were selected and assessed in binding assays.

Peptides were screened for binding to recombinant human HLA-E*01:03 using a thermal shift assay, which allows the determination of the thermal melting point (Tm) of the peptide–HLA-E complex (pHLA-E) in the presence of excess peptide[36], providing a proxy for the complex stability. Only 6 of 70 peptides tested showed a measurable Tm value, with the peptide ILSPFLPLL (Env$_{371-379}$) demonstrating the highest thermal stability (Tm value of 47.1 °C) (Fig. 1a). This peptide is present in all forms of the HBV Env proteins (large, middle, and small). A known HLA-E binder peptide from *Mycobacterium tuberculosis* (Mtb) (RLPAKAPLL, herein referred to as Mtb RLPA) was used as a positive control and exhibited a Tm value of 51.4 °C (Fig. 1a)[31].

To further investigate the binding capacity of Env$_{371-379}$ peptide to both *HLA-E* alleles, a cell pulsing assay was performed. The Env$_{371-379}$ peptide plus the top six most conserved genotype variants of this peptide were tested alongside the Mtb RLPA peptide and three signal peptides derived from the leader regions of three HLA alleles. HLA class I-deficient K562 cells were transduced with $\beta_2$m-HLA-E*01:01 or $\beta_2$m-HLA-E*01:03 single chain gene-fusion constructs to generate K562-E*01:01 and K562-E*01:03 cells, respectively. These cells were pulsed separately with each peptide at a final concentration of 100 μg/mL for 2 h. The level of HLA-E on the cell surface was then measured by flow cytometry. Cells pulsed with the signal peptides, Mtb RLPA peptide, and five out of six variants of Env$_{371-379}$ peptide showed increased cell surface HLA-E expression, indicating peptide binding and stabilization of HLA-E molecules on the cell membrane. The exception was the triple variant peptide Env$_{371-379}$ (L2V-S3R-L6I), which did not show an increase in HLA-E expression compared to the unpulsed cells (Fig. 1b and Supplementary Fig. 1).

The Env$_{371-379}$ peptide and the two variants eliciting the largest HLA-E surface upregulation (Env$_{371-379}$, S3N and Env$_{371-379}$, L6I) were selected for further analysis. The t$_{1/2}$ of Env$_{371-379}$, Env$_{371-379}$ (S3N), and Env$_{371-379}$ (L6I) pHLA-E*01:03 complexes was measured using surface plasmon resonance (SPR) by monitoring binding over time of a solu-bilized HLA-specific receptor, ILT2, to functionally refolded monomers[37]. With a t$_{1/2}$ of 2 hours, the Env$_{371-379}$ (L6I) peptide showed the greatest capacity to stabilize HLA-E on the cell surface, followed by Env$_{371-379}$ (S3N) (with t$_{1/2}$ of 39.5 min) and the Env$_{371-379}$ peptide (t$_{1/2}$ of 6.7 min) (Fig. 1b, c). The t$_{1/2}$ of pHLA-E*01:03 complexes of two signal peptides (A1 and Cw3) was also measured for comparison, and both peptides showed a t$_{1/2}$ of 2 h (Fig. 1c). Next, the thermal melting point (Tm) of the same pHLA-E*01:03 complexes were measured using a thermal shift assay. Env$_{371-379}$ (L6I) and the two signal peptides showed a similar Tm of 49 °C, Env$_{371-379}$ (S3N) had a Tm of 47 °C and Env$_{371-379}$ peptide a Tm of 44 °C (Fig. 1c). The Tm and t$_{1/2}$ of HLA-E*01:03 molecules refolded in the absence of peptide could not be determined due to the instability of these molecules.

Collectively, these data indicated that the Env$_{371-379}$ peptide and two of its variants bind HLA-E with different stabilities. To understand

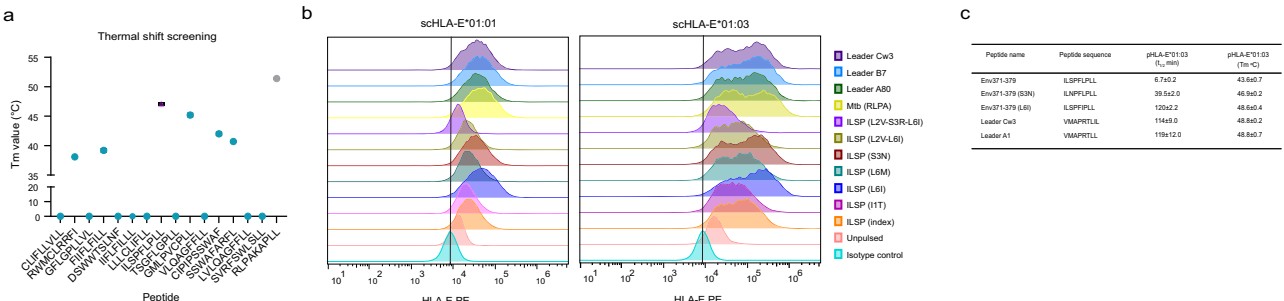

**Fig. 1 | Identification and validation of HBV peptides binding HLA-E. a** Thermal melting point (Tm) assessed by thermal shift assay screen of HBV peptides (Env$_{371-379}$ peptide, magenta triangle) and Mtb RLPA (positive control, gray circle). Peptides were added to refolded and purified HIV Gag$_{275-283}$-HLA-E*01:03 complexes (in PBS at 0.25 mg/mL) at a 60:1 molar ratio. Data were presented as mean values ± SD of triplicates. **b** HLA-E binding of HBV Env$_{371-379}$ variants, Mtb RLPA, and signal peptides (A80, B7, and Cw3) was measured by HLA-E upregulation at the cell surface of K562 cells transduced with single chain β$_2$m-HLA-*E*01:01 or β$_2$m-HLA-E*01:03*. Plotted are flow cytometry histograms after staining with 3D12 antibody (HLA-E antibody). Histograms are representative of two independent experiments, each conducted in triplicates (full dataset and gating strategy available in Supplementary Fig. 1). **c** Summary of t$_{1/2}$ and Tm of HBV Env$_{371-379}$ variants and signal peptides (Cw3 and A1) pHLA-E*01:03 complexes as determined by SPR and thermal shift assays in triplicate. All SPR experiments were performed at 25 °C. Data were presented as mean values ± SD of triplicates. Source data are provided as a Source Data file.

the potential therapeutic relevance of our findings, we determined the genotype prevalence of the HBV Env$_{371-379}$ peptide variants across the five genotypes A–E. We calculated that the Env$_{371-379}$ peptide is the most prevalent in genotypes C and D (65 and 84% of the sequences analysed, respectively), while Env$_{371-379}$ (S3N) is present in genotype C (21%) and Env$_{371-379}$ (L6I) is most prevalent in genotypes A and E (45 and 91%, respectively) (Supplementary Table 1).

## Generation of a TCR targeting three HLA-E HBV Env$_{371-379}$ peptides

To assess the importance of HLA-E complex stability for efficient epitope presentation, we generated a TCR capable of recognizing the HLA-E molecules complexed with all three Env$_{371-379}$ peptide variants. Naïve TCR phage libraries were screened, with one TCR selected for further development due to its micromolar binding affinity to Env$_{371-379}$ (S3N), Env$_{371-379}$ (L6I) as well as Env$_{371-379}$ peptide–HLA-E complexes (Fig. 2a and Supplementary Fig. 2a). By randomizing the CDR loops coupled with phage display selection, the affinity of the wild-type TCR to all three target complexes was enhanced around one million-fold. To ensure specificity to the target, phage libraries were depleted with a mixture of signal peptide pHLA-E complexes prior to positive selection with HBV pHLA-E complexes. This procedure resulted in a TCR mutant (a09b08) with binding affinity to all three HBV pHLA-E complexes in the picomolar range (4.5–11 pM) and half-life of 20 h (Fig. 2b, c and Supplementary Fig. 2b). As additional specificity controls, we showed that the TCR mutant a09b08 did not bind to any of the three HBV pHLA-A*02:01 complexes, despite known presentation of Env$_{371-379}$ by HLA-A*02:01 molecules[38] (Supplementary Fig. 3) nor to any of the seven signal peptide pHLA-E complexes tested (Supplementary Table 2).

The crystal structures of the a09b08 TCR in complex with all three HBV pHLA-E*01:03 complexes were solved at 2.25, 2.35, and 2.61 Å resolution for the Env$_{371-379}$, Env$_{371-379}$ (S3N) and Env$_{371-379}$ (L6I) peptides, respectively (Supplementary Table 3). In all three structures, the a09b08 TCR engages with pHLA complexes in the same way, and in a canonical fashion, with the TCR alpha chain positioned over HLA helix 2 and the TCR beta chain positioned over HLA helix 1, allowing both CDR3s to sit centrally above the HLA peptide-binding groove and interact with the peptide (Fig. 3a). The TCR binds pHLA-E with a crossing angle of 46° and burying an average surface area of 303 Å² on peptide and 848 Å² on HLA. Unambiguous electron density at position 3 for S3N and position 6 for L6I enabled the modeling of the peptide variants (Supplementary Fig. 4a–c). The three HBV peptide variants adopt a highly similar conformation, and the positions of residues in the HLA-E binding groove were also largely unchanged (Fig. 3b, c and

Supplementary Fig. 5a). The three HBV peptides also adopt a broadly similar binding conformation to previously described HLA-E binders, Mtb RLPA peptide and canonical VL9 signal peptide[39,40] (Supplementary Fig. 6). The Env$_{371-379}$ (S3N) variant makes an additional hydrogen bond to HLA Q156 in helix 2 that is not present with the other two peptides (distance cut-off of 3 Å), consistent with the stability increase seen with Env$_{371-379}$ (S3N) compared to the Env$_{371-379}$ peptide (Supplementary Fig. 5b). Structural overlay of Env$_{371-379}$ and Env$_{371-379}$ (L6I) variants showed minor differences surrounding peptide position 6, with the I6 backbone slightly moved towards the HLA helix 1 and concurrently HLA residues T70 and F74 pulled towards the Env$_{371-379}$ (L6I) peptide. Additionally, the orientation of the peptide I6 side chain enabled interaction with HLA F74, which is further apart in the case of L6 (Supplementary Fig. 5c). The peptide contacts are predominantly through hydrophobic interactions, including the TCR alpha chain H94 stacking with I1 and P4, multiple TCR residues stacking with the side chain of F5 which protrudes from the binding groove, and the TCR beta chain R95 with L8. Additionally, the side chain of the TCR beta chain N96 (one of the residues introduced during affinity maturation) forms hydrogen bonds with the peptide backbone at position 6 (Fig. 3b and Supplementary Table 4). Overall, the interactions between the a09b08 TCR and the HLA-E heavy chain were highly similar for all three peptides, with limited variation observed due to differences in side-chain conformations of residues (Supplementary Table 4). Finally, the TCR residues interacting with HLA-E or the peptide remained the same between the three complexes (Fig. 3d, e).

## The a09b08 ImmTAV molecule redirects T cells against antigen$^+$ targets

The a09b08 TCR was converted to an ImmTAV molecule by the addition of a single chain variable fragment domain (scFv) anti-CD3 arm for effector function, thereby generating a tool to monitor HBV peptide presentation in cellular assays. As target cells, we first used THP-1 cells deficient in β$_2$m and *CIITA* genes (thus lacking both classical HLA class -I and -II molecules) and transduced with a single chain gene-fusion construct encoding for HLA-E*01:01 or HLA-E*01:03 and β$_2$M proteins (herein referred to as THP-1-E cells, Supplementary Fig. 7a, c). PBMC from HBV-naïve donors secreted IFN-γ when redirected by the a09b08 ImmTAV molecule against THP-1-E cells expressing either HLA-E allele and pulsed with the Env$_{371-379}$, Env$_{371-379}$ (S3N) or Env$_{371-379}$ (L6I) peptides, although with different dose-response curves. In agreement with the t$_{1/2}$ of the complexes (Fig. 1c), we observed lower ImmTAV EC$_{50}$ values against THP-1-E*01:03 cells pulsed with Env$_{371-379}$ (L6I) peptide (3.97 pM), followed by Env$_{371-379}$ (S3N) peptide (5.34 pM)

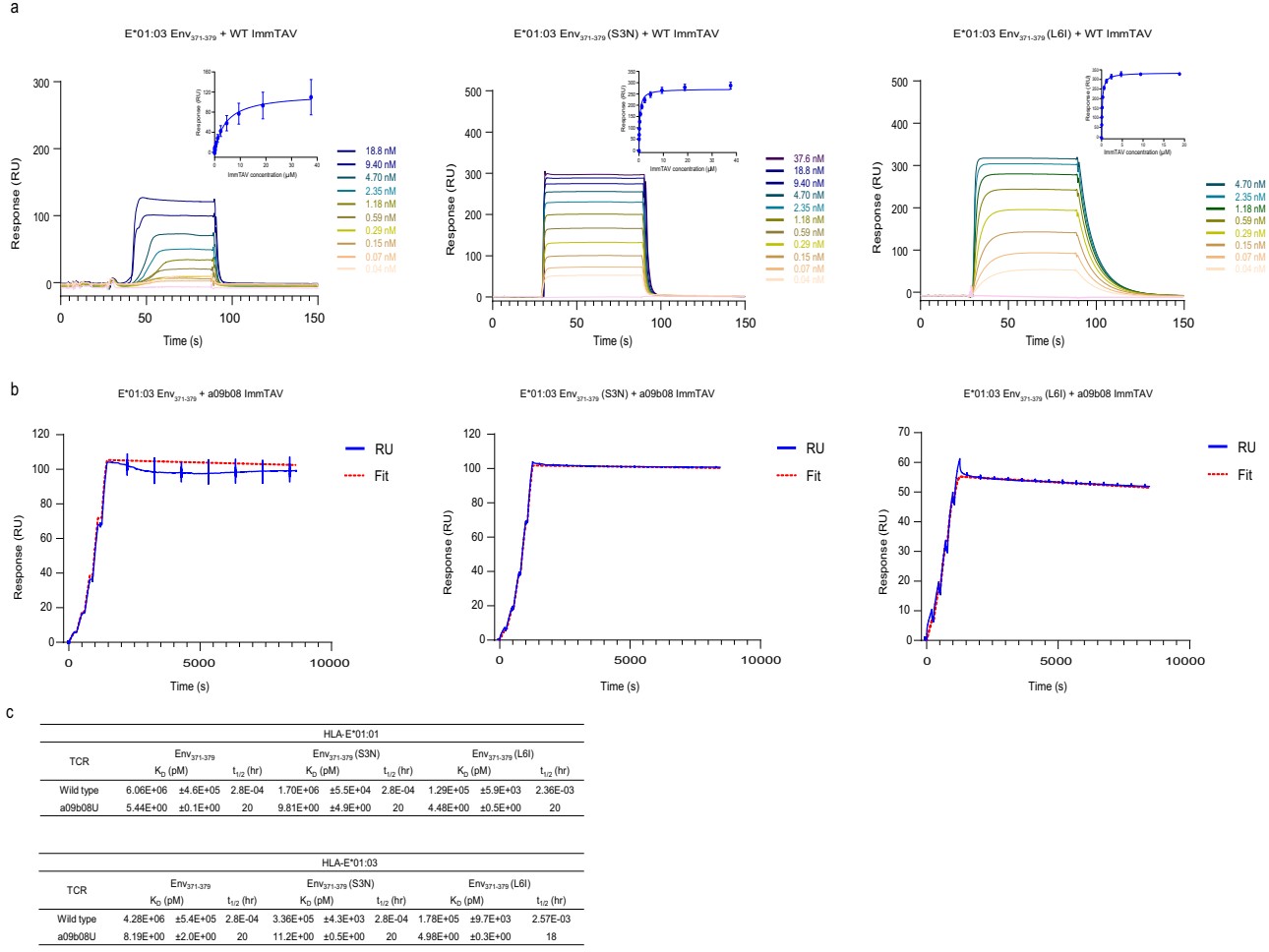

**Fig. 2 | Binding kinetics of wildtype and a09b08 ImmTAVs to HBV Env$_{371-379}$ pHLA-E*01:03 complexes. a** Binding curves for wildtype ImmTAV and all three HBV Env$_{371-379}$ pHLA-E*01:03 complexes. Binding was determined over a range of analyte concentrations from 37 nM to 37.6 μM. Insets: calculation of steady-state affinity, data were presented as mean values ± SD. **b** Binding kinetics for a09b08 ImmTAV and all three HBV Env$_{371-379}$ pHLA-E*01:03 complexes. Graphs show the mean of the raw data (blue) and the 1:1 fit (dotted red line). For the binding kinetic graphs, the ImmTAV molecule was flown over the chip as the analyte, at concentrations ranging from 0.313 to 5 nM. Kinetic constants were determined using a 1:1 Langmuir model. **c** Summary of $K_D$ values and t$_{1/2}$ values of the wild-type and a09b08 ImmTAV. All experiments were performed at 25 °C in triplicate. Data were presented as mean values ± SD. Source data are provided as a Source Data file.

and Env$_{371-379}$ peptide (60.9 pM) (Fig. 4a and Supplementary Table 5). The same hierarchy was observed with peptide-pulsed THP-1-E*01:01 cells as targets, but with higher EC$_{50}$ values (Supplementary Fig. 8a and Supplementary Table 5). Peptide titration experiments showed that responses were detected even at low Env$_{371-379}$ (L6I) peptide concentrations (1 and 700 ng/mL for THP-1-E*01:03 and THP-1-E*01:01 respectively, Fig. 4b, Supplementary Fig. 8b, and Supplementary Table 5). Intermediate peptide concentrations were required to detect responses to the Env$_{371-379}$ (S3N) peptide (0.2 and 4.9 μg/mL for THP-1-E*01:03 and THP-1-E*01:01 respectively), while responses to Env$_{371-379}$ peptide were detected only at high peptide concentrations (8.7 μg/mL for THP-1-E*01:03) (Fig. 4b, Supplementary Fig. 8b, and Supplementary Table 5). As the a09b08 ImmTAV molecule has similar binding kinetics to all three pHLA-E complexes (Fig. 2c), the differential potency observed in cellular assays reflects the stability and overall cell surface presentation of the complexes (Fig. 1b, c).

To assess the endogenous presentation of HLA-E peptides, HepG2 target cells were stably transfected with minigenes encoding the HBV Env$_{371-379}$, Env$_{371-379}$ (S3N) or Env$_{371-379}$ (L6I) peptides. The a09b08 ImmTAV molecule induced specific IFN-γ release from HBV-naïve donor PBMC in the presence of HepG2 presenting the Env$_{371-379}$ (S3N) and Env$_{371-379}$ (L6I) peptides, but not the Env$_{371-379}$ peptide or non-transduced HepG2 cells (Fig. 4c and Supplementary Table 5). ImmTAV-

dependent IFN-γ release against HepG2 targets transfected with minigenes was inhibited by the anti-HLA-E blocking mAb 3D12, but not by an anti-HLA-A2 antibody, despite HepG2 being HLA-A*02:01$^+$ (Fig. 4d and Supplementary Table 6), confirming HLA-E-restriction and specificity of the a09b08 ImmTAV molecule. No cross-reactivity of the a09b08 ImmTAV molecule was detected when THP-1-E cells were separately pulsed with various signal peptides derived from different HLA alleles, two putative mimetic peptides from the human proteome (defined in Supplementary Methods) and known HLA-E peptide ligands from self or other microbes (Supplementary Table 7). Together these results provide evidence for the selectivity of the a09b08 ImmTAV for HLA-E-Env$_{371-379}$ complexes and highlight the importance of the pHLA-E complex stability for TCR recognition of endogenously derived peptides.

**The a09b08 ImmTAV molecule mediates the HLA-E-dependent killing of targets**
To determine whether the a09b08 ImmTAV molecule could also elicit T cell killing of targets via recognition of endogenously presented Env$_{371-379}$ (L6I), PBMC were cocultured with PLC/PRF/5 (Ag$^+$), an HCC cell line that expresses HBsAg from integrated HBV DNA (Supplementary Fig. 9)[41–44]. Untransduced or HLA-E*01:03-transduced PLC/PRF/5 cells (Supplementary Fig. 7b, d) were incubated with HBV-naïve

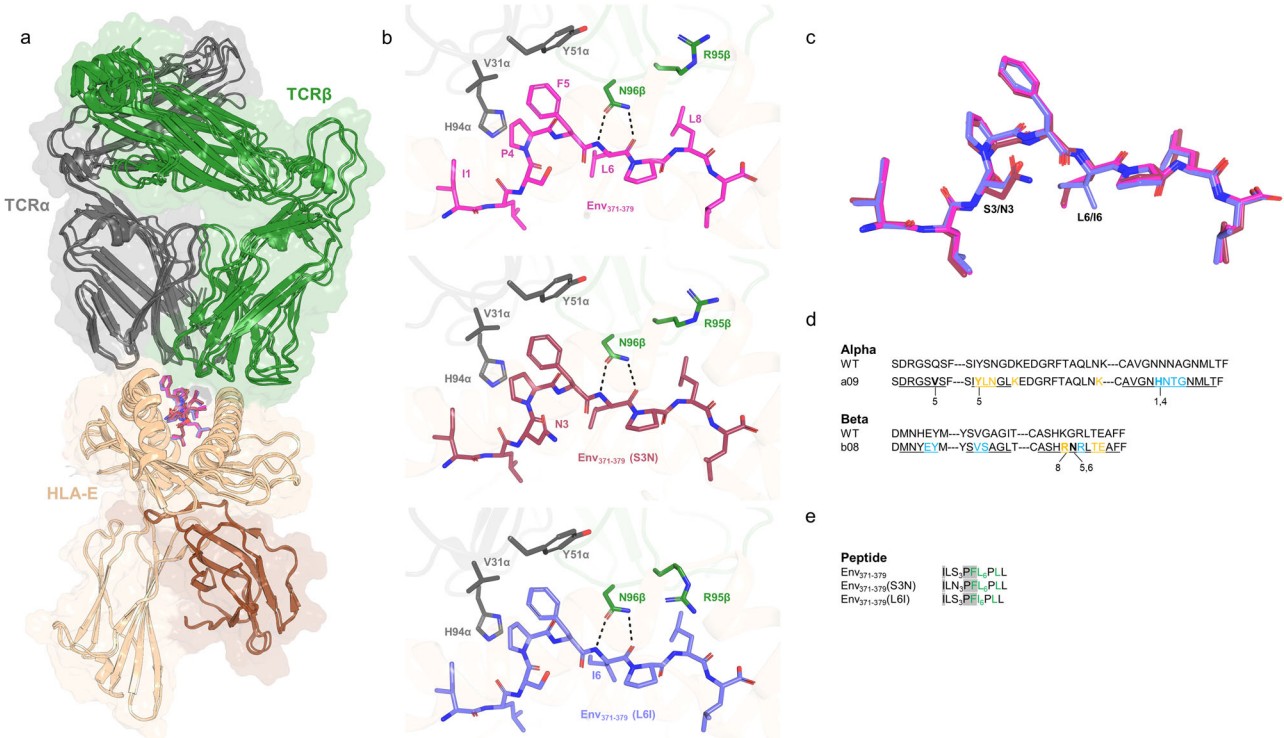

**Fig. 3 | Structural overview of the a09b08 TCR in complex with three HBV Env₃₇₁₋₃₇₉ pHLA-E complexes. a** Cartoon overview of the three TCR-pHLA complexes aligned on HLA-E*01:03 (TCR alpha chain in gray, TCR beta chain in green, HLA-E in wheat, B2M is brown, peptides: Env₃₇₁₋₃₇₉ in magenta, Env₃₇₁₋₃₇₉ (S3N) in red and Env₃₇₁₋₃₇₉ (L6I) in blue). **b** Close-up view of the three TCR-peptide interactions. TCR residues within 4 Å of the peptide are shown as sticks. Dotted lines indicate polar contacts. **c** Overlay of the three peptides, aligned on HLA-E, showing highly similar conformation in the complexes. **d** a09b08 TCR contacts to pHLA-E mapped onto the truncated TCR sequence. A wild-type TCR sequence is provided to show the mutations introduced by affinity maturation. CDR residues are underlined and residues within 4.1 Å of the peptide are highlighted in bold with peptide positions indicated below. Residues highlighted in blue and orange indicate positions that are within 4.1 Å of the HLA-E helix 1 and HLA-E helix 2, respectively. **e** a09b08 TCR contacts to peptide mapped onto the peptide sequences. Peptide residues within 4.1 Å of the TCR alpha chain are highlighted gray, and those of the TCR beta chain are colored green.

PBMC in the presence of ImmTAV for 3 days, and caspase-3/7 activation was used to measure cell death. Target cells were lysed in a dose-dependent manner in the presence of ImmTAV, despite the peptide being presented at levels close to or below the limit of detection by targeted tandem mass spectrometry (Supplementary Fig. 10). Target cell lysis was detected from 27 h of coculture, and maximum cytolysis was achieved at 0.1 and 1 nM ImmTAV concentrations by 72 h (Fig. 5a). Faster lysis kinetics (target cell lysis from 18 h) were observed against HLA-E*01:03-transduced target cells (Fig. 5b). These results demonstrate the sensitivity of the a09b08 ImmTAV to very low levels of HLA-E-Env₃₇₁₋₃₇₉ (L6I) complexes.

**The a09b08 ImmTAV elicits activation of redirected effector T cells**

To confirm that the a09b08 ImmTAV could redirect T cells against HBV-transfected targets, we transiently transfected HepG2 cells with replication-competent infectious 1.3-mer cDNA clones of HBV genotypes expressing the three Env₃₇₁₋₃₇₉ epitopes[45]. As HepG2 cells are homozygous for *HLA-E*01:01* and the cell surface expression of HLA-E is very low under basal conditions (Fig. 6a and Supplementary Table 6), we pretreated them with IFN-γ (to increase HLA-E expression, Fig. 6a) or transduced them with lentiviral particles encoding *HLA-E*01:03* (herein referred to as HepG2-E cells) (Fig. 6a and Supplementary Fig. 11). Transfection efficiency of HepG2 cells with HBV cDNA clones was determined by quantification of the cell population positive for intracellular HBsAg expression 24 h post-transfection (Supplementary Fig. 12). Untransfected and transfected HepG2 cells were cocultured for 72 h with CD3⁺ T cells alone or in the presence of

a09b08 ImmTAV at 1 or 10 nM concentrations (Supplementary Fig. 13a). T cell activation was determined by measuring the surface expression of activation markers (CD25 and CD69) by flow cytometry. In the presence of the a09b08 ImmTAV molecule, CD4 and CD8 T cell activation was observed only in co-cultures with IFN-γ-pretreated wildtype HepG2 cells or HepG2-E cells transfected with HBV-Genotype A2 (which expresses Env₃₇₁₋₃₇₉ L6I) (Supplementary Fig. 13b, c). No T cell activation was elicited against HepG2 targets transfected with Env₃₇₁₋₃₇₉ (S3N)-Genotype C1 or Env₃₇₁₋₃₇₉-Genotype C2 (Supplementary Fig. 13b, c).

To determine the efficacy of the a09b08 ImmTAV in mediating T cell cytotoxicity, we measured IFN-γ and granzyme B (GzmB) levels in the culture supernatant collected at days 3 and 4 post coculture. A dose-dependent release of IFN-γ and GzmB by T cells was detected in the presence of the a09b08 ImmTAV molecule in co-cultures with IFN-γ-pretreated HepG2 or HepG2-E cells transfected with Env₃₇₁₋₃₇₉ (L6I)-Genotype A2 (Fig. 6b−d and Supplementary Fig. 13d, e). No cytokines were elicited by an irrelevant Immune mobilizing monoclonal T cell receptors against bacteria (ImmTAB), targeting HLA-E Mtb RLPA peptide complexes[46] (Fig. 6c, d). As an additional control, we included the HBV HLA-A*02:01 ImmTAV (targeting a peptide within the HBV envelope protein[17]). HLA-E-a09b08 ImmTAV and HLA-A*02:01 ImmTAV molecules elicited comparable IFN-γ and GzmB secretion against targets overexpressing *HLA-E*01:03*, however only the HLA-A*02:01 ImmTAV molecule redirected T cells to HBV-transfected wildtype HepG2 cells (Fig. 6c, d). The HLA-A*02:01 ImmTAV molecule also elicited fivefold greater cytokine responses against IFN-γ−treated HBV-transfected HepG2 cells compared to HLA-E-a09b08 ImmTAV (Fig. 6c, d). Altogether, these results demonstrate that the surface

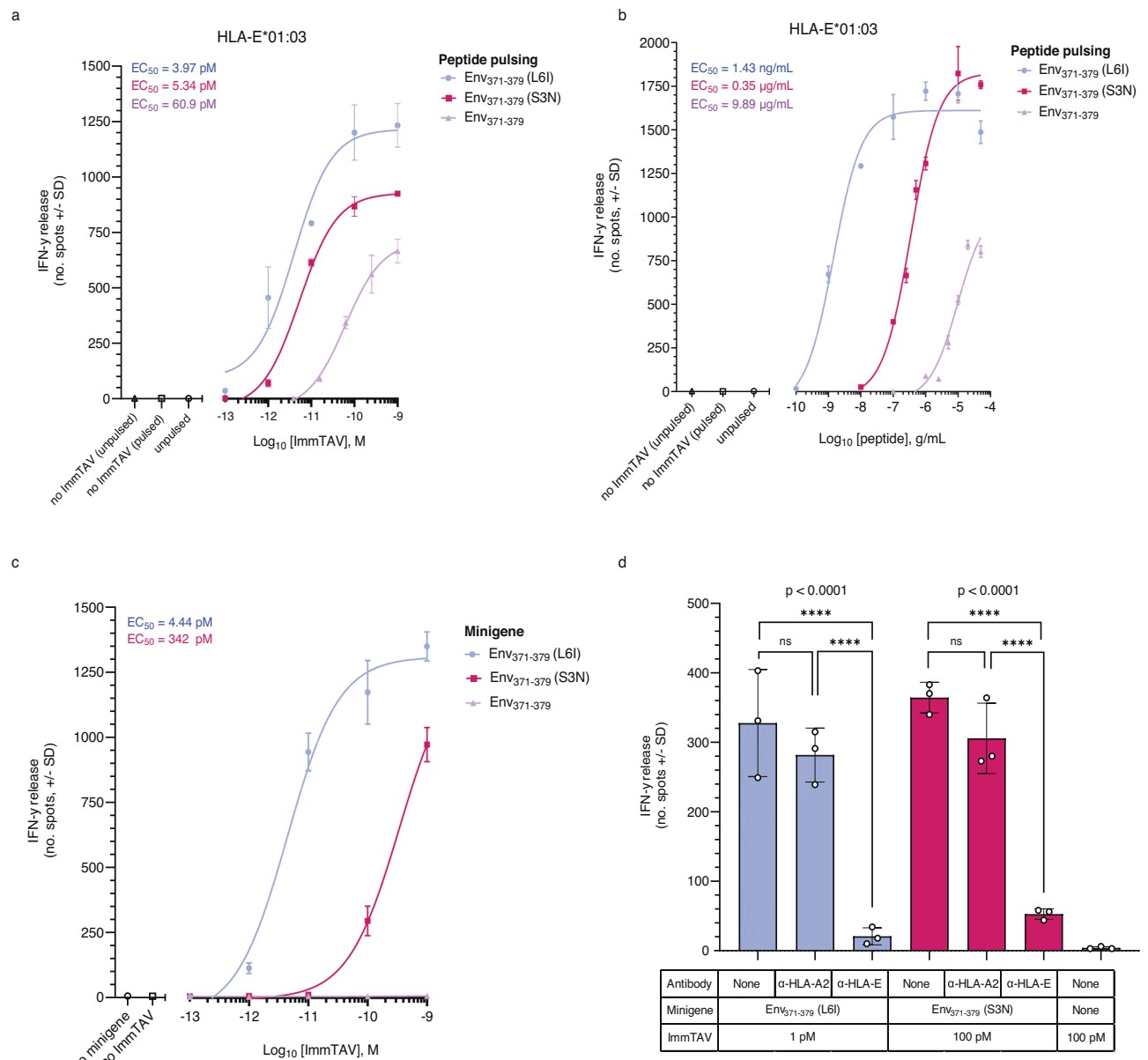

**Fig. 4 | The a09b08 ImmTAV molecule elicits T cell responses against cells displaying HBV Env$_{371-379}$ pHLA-E complexes. a** ELISPOT assay measuring dose-dependent IFN-γ release induced by a09b08 ImmTAV in co-cultures of PBMC from 3 HBV-naïve donors and THP-1-E*01:03 cells pulsed with Env$_{371-379}$ (purple line), Env$_{371-379}$ (S3N) (red line) and Env$_{371-379}$ (L6I) (blue line) peptides (10 μg/mL). **b** IFN-γ ELISPOT assays showing titratable activation of PBMC from 3 HBV-naïve donors by a09b08 ImmTAV (1 nM) in the presence of THP-1-E*01:03 cells pulsed with the indicated amounts of peptide. Controls (**a**, **b** clear symbols) include PBMC + target cells (no ImmTAV; unpulsed or pulsed with 10 μg/mL peptide), and PBMC + ImmTAV + target cells (unpulsed). **c** ELISPOT assay measuring dose-dependent IFN-γ release from HBV-naïve donor PBMC in the presence of HepG2 targets stably

transfected with minigenes encoding the indicated peptides Env$_{371-379}$, Env$_{371-379}$ (S3N), and Env$_{371-379}$ (L6I). Untransfected HepG2 (no minigene) and PBMC + minigene target cells (no ImmTAV) were included as controls (clear symbols). **d** Cumulative IFN-γ responses induced by a09b08 ImmTAV in PBMC cocultured with HepG2 Env$_{371-379}$ (L6I) (blue) or Env$_{371-379}$ (S3N) (red) minigene targets in the presence or absence of blocking mAbs against HLA-E or HLA-A*02:01 (10 μg/mL); ****$p$ = <0.0001, two-way ANOVA. Results (**a**–**d**) are representative of one of three PBMC donors tested in triplicate. Data were presented as mean values ± SD. Average EC$_{50}$ are indicated in the inset at the top left corner of the figures (**a**–**c**). All donor EC$_{50}$ values and averages are displayed in Supplementary Table 5. Source data are provided as a Source Data file.

density of pHLA-E complexes is a key determinant of the strength of the response.

## The a09b08 ImmTAV mediates inhibition of viral replication in vitro

To evaluate whether a09b08 ImmTAV-redirected T cells could inhibit viral replication, we quantified HBeAg and HBsAg levels in the supernatant of the assay described in Fig. 6b, at day 4 and 6 post coculture. A significant decrease of HBeAg and HBsAg levels in the presence of a09b08 ImmTAV and HBV HLA-A*02:01 ImmTAV molecules was

observed in co-cultures with IFN-γ-pretreated wildtype HepG2 or HepG2-E cells (Fig. 6e–h). However, the a09b08 ImmTAV molecule did not induce anti-viral activity against wild-type HepG2 cells, in contrast to the HBV HLA-A*02:01 ImmTAV molecule (Fig. 6e–h). The control, irrelevant HLA-E ImmTAB Mtb RLPA, did not elicit any anti-viral activity (Fig. 6e–h). Also, no anti-viral activity was elicited by a09b08 ImmTAV against HepG2 targets transfected with Env$_{371-379}$ (S3N)-Genotype C1 or Env$_{371-379}$-Genotype C2 (Supplementary Fig. 13f, g). The HLA-A*02:01 ImmTAV molecule also elicited ~5-fold greater anti-viral activity against IFN-γ treated HBV-transfected HepG2 cells compared to the HLA-E-

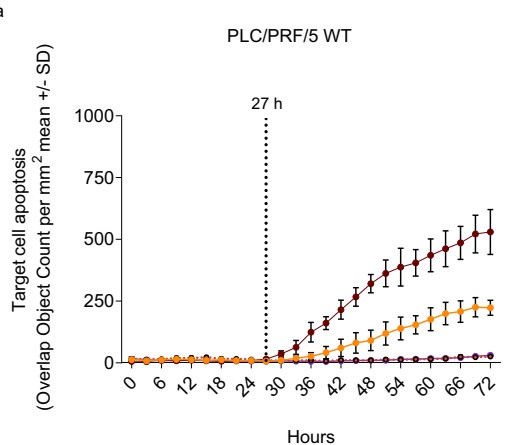

**Fig. 5 | The a09b08 ImmTAV elicits antigen-dependent T cell killing of target cells.** Apoptosis (measured by IncuCyte assay, see methods) of Env$_{371-379}$ (L6I) Ag$^+$ PLC/PRF/5 target cells wildtype (**a**) or HLA-E*01:03 lentiviral transduced (**b**), cocultured with HBV-naïve donor PBMC and a titration of a09b08 ImmTAV.

Dashed vertical lines indicate timepoints from which target cell apoptosis begins (at 1 nM ImmTAV). Results (**a**, **b**) are representative of one of three PBMC donors, each tested in triplicate. Data were presented as mean values ± SD. Source data are provided as a Source Data file.

a09b08 ImmTAV (Fig. 6e–h), confirming that density and stability of pHLA-E complexes are critical for effector T cell redirection with our bispecific molecule.

### Detection of HLA-E Env$_{371-379}$ specific CD8$^+$ T cells in peripheral blood

Having demonstrated that HBV-transfected cells can present Env$_{371-379}$ (L6I) pHLA-E complexes at sufficient levels to trigger T cell activation, the killing of infected targets, and antiviral activity, we next investigated whether we could detect HLA-E-restricted CD8$^+$ T cells specific for the HBV Env$_{371-379}$ (L6I) epitope in CHB donors. People with CHB were selected according to the inclusion criteria described in Supplementary Methods. Three of the donors were HBsAg-negative at the time of sampling despite confirmed CHB, suggesting that they had resolved infection (Supplementary Table 8). As determined by sequencing, four CHB donors carried the Env$_{371-379}$ (L6I) sequence, three carried the Env$_{371-379}$ sequence, and one carried the Env$_{371-379}$ (S3N) variant, with the remaining carrying less prevalent variants (Supplementary Table 8). PBMC samples were analysed by flow cytometry using fluorescent pHLA-E dextramers. HBV-naïve donor PBMC were first spiked with Jurkat cells transduced with the wild-type TCR to establish the detection limit of our dextramer-E reagent, which was 0.001%, or 10 cells per million PBMC (Supplementary Fig. 14a). HBV-naïve donor PBMC showed no background staining with dextramer-E (Supplementary Fig. 14a). As an additional dextramer-E specificity control, we showed that TCR-transduced Jurkat clones did not stain with either the HLA-A*02:01 Env$_{371-379}$ (L6I) dextramer or the HLA-E*01:03 Cw3 signal peptide dextramer (Supplementary Fig. 14b).

No HBV-specific HLA-E-restricted CD8$^+$ T cells were detected in HBV-naïve or CHB donor PBMC after ex vivo staining with dextramer-E (Supplementary Figs. 15, 16). However, HBV-specific HLA-E-restricted CD8$^+$ T cells were detected by dextramer staining in seven out of ten CHB donor samples and in one out of five HBV-naïve donors (Fig. 7 and Supplementary Fig. 16), following two rounds of in vitro stimulations with artificial antigen-presenting cells (refolded Env$_{371-379}$ (L6I)-pHLA-E conjugated to magnetic beads and anti-CD28 antibody, capable of activating the Jurkat clones, Supplementary Fig. 14). In conclusion, although Env$_{371-379}$ (L6I)-HLA-E restricted T cells were not detectable ex vivo in either CHB or HBV-naïve donor PBMC, possibly because they may not be present in the circulation, these results show that they could be expanded after in vitro priming and at a higher frequency in the former, suggestive of prior expansion in vivo as a result of naturally occurring presentation of this epitope.

## Discussion

In this study, we identified an HLA-E-restricted HBV-derived epitope, Env$_{371-379}$ ILSPFLPLL. We observed that the Env$_{371-379}$ peptide sequence forms pHLA-E complexes with limited stability (t$_{1/2}$ of 6.7 min), while two peptide variants, Env$_{371-379}$ (S3N) and Env$_{371-379}$ (L6I), bind HLA-E with greater stability, reaching t$_{1/2}$ of 39.5 min and 2 h, respectively. In agreement with the known effect on the surface pHLA-E stability of the single amino acid difference (at position 107) between HLA-E*01:01 and HLA-E*01:03[47], we also observed higher stability of HLA-E*01:03 Env$_{371-379}$ complexes, irrespective of the genotype variant. These results are also in agreement with the preference of HLA-E for binding HLA class I signal sequence peptides, with substitutions beyond the canonical residues known to destabilize the complex and its display on the cell surface[31,39].

To investigate the presentation of the identified HBV peptides in the context of HLA-E, we developed an ImmTAV molecule with picomolar affinity for HLA-E-Env$_{371-379}$ complexes. The a09b08 ImmTAV molecule specifically and potently targeted all three HLA-E-presented HBV Env$_{371-379}$ peptide variants in T cell redirection assays with HBV Env-positive targets. These desirable characteristics were achieved by introducing multiple sequence modifications in the TCR CDR regions during the engineering process, thereby generating novel interactions with both the peptide and HLA-E, as shown by structural analyses of the trimolecular peptide−HLA-E-TCR complexes. Mutations in the three variants significantly influenced peptide−HLA-E stability but had no impact on TCR recognition. Of note, our structural analysis did not explain the differences observed in peptide−HLA-E stability across the HBV Env$_{371-379}$ peptide variants. Certain T cell epitopes can be presented by both HLA-A*02:01 and HLA-E molecules[48,49], and a 9-mer version of the HBV Env peptide was shown to be presented by HLA-A*02:01 molecules in humans[38], therefore we considered the possibility that our a09b08 ImmTAV molecule could be cross-reactive to the Env$_{371-379}$ peptide in the context of HLA-A*02:01. However, we demonstrated that the a09b08 ImmTAV is specific for HLA-E Env$_{371-379}$ peptide complexes, with no cross-reactivity for HLA-A2 molecules refolded with HBV Env$_{371-379}$. Additionally, we did not observe the cross-reactivity of the a09b08 ImmTAV molecule to HLA-E-leader peptide complexes, either by SPR or in cellular assays. However, further systematic screening would need to be conducted prior to consideration for a first-in-human clinical trial.

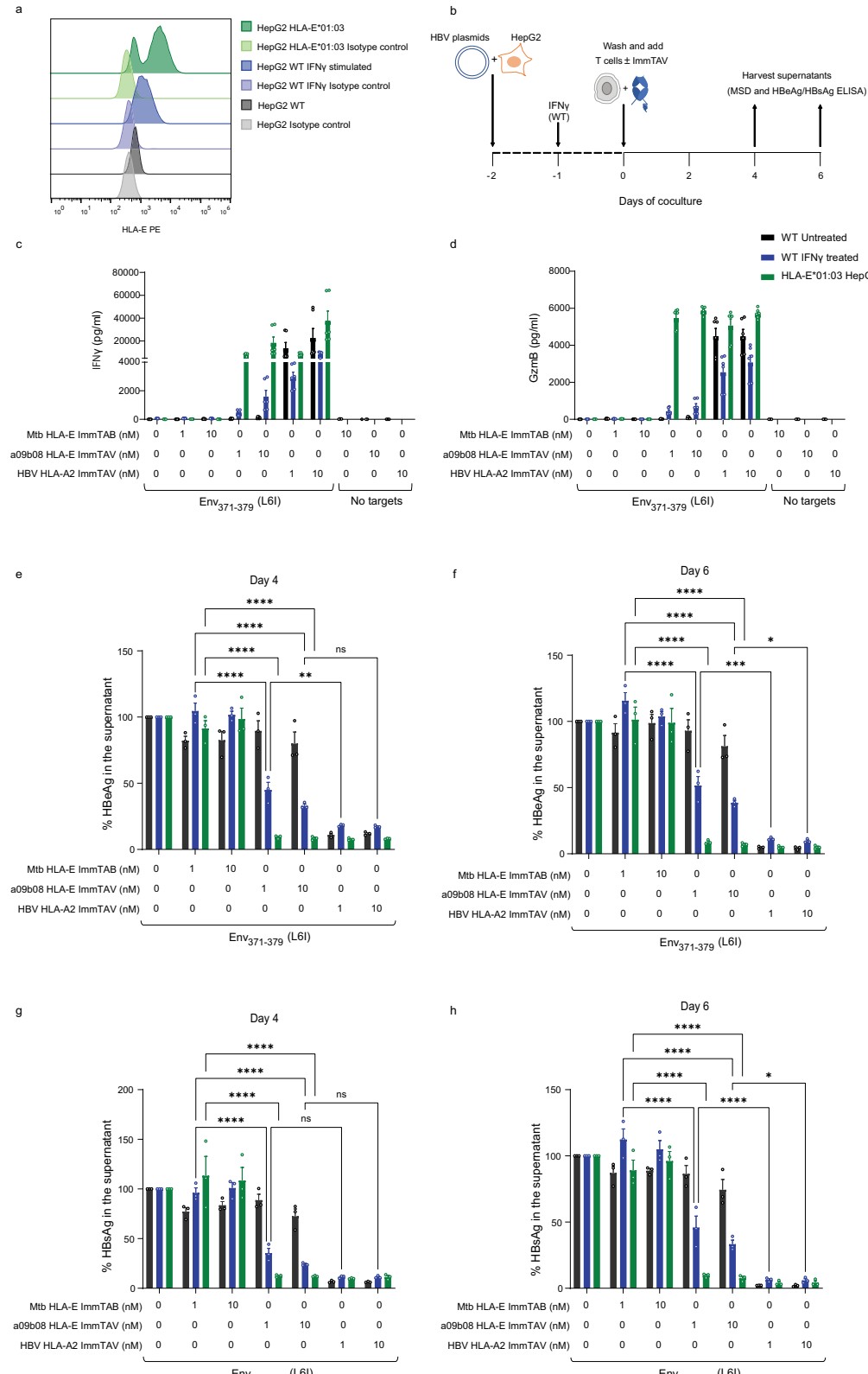

In agreement with the stability of refolded HBV pHLA-E complexes, we observed that the a09b08 ImmTAV molecule was most effective in redirecting T cells against THP-1-E cells pulsed with Env$_{371-379}$ (S3N) and Env$_{371-379}$ (L6I) variant peptides. Furthermore, recognition of the non-mutated Env$_{371-379}$ peptide was completely lost when endogenously expressed from a minigene in HepG2 cells.

This loss of recognition could be attributed to the low affinity of the Env$_{371-379}$ peptide for HLA-E, thereby hindering its presentation on HLA-E at the cell surface, and/or the potential sequestration of the Env$_{371-379}$ peptide by HLA-A2 molecules in the context of assays with HepG2 cells as targets. The a09b08 ImmTAV molecule was still able to redirect T cells against HepG2 targets transfected with minigenes

**Fig. 6 | The a09b08 ImmTAV activates T cells to eliminate HBV-transfected HepG2 cells. a** Surface expression levels of HLA-E on HepG2 cells were analysed by flow cytometry. HepG2 wildtype and HLA-E*01:03 overexpressing cells were transfected with a replication-competent HBV plasmid encoding genotype A2 (Env$_{371-379}$ (L6I)). **b** 24 h post-transfection, some wells of HepG2 wildtype cells were stimulated with IFN-γ (1 ng/mL) for 16 h. Cells were washed and cocultured with pan T cells from three donors (1:1 ratio) with or without a09b08 ImmTAV, HBV HLA-A*02:01 ImmTAV or Mtb RLPA HLA-E ImmTAB (negative control) at 1 and 10 nM. Culture supernatants were harvested on day 4 and day 6 according to the schematics shown (modified from ref. 17). Panel **b** is released under a Creative

Commons Attribution-Non Commercial 4.0 International license (https://creativecommons.org/licenses/by-nc/4.0/deed.en). Levels of IFN-γ (**c**) and granzyme B (**d**) in the culture supernatants at day 4 were quantified using MSD. Data represents the mean ± SEM of triplicates (*n* = 3). HBeAg (**e**, **f**) and HBsAg (**g**, **h**) levels in the culture supernatant at day 4 and day 6 were quantified using ELISA. Data were represented as mean ± SEM of duplicates from three donors. Significant differences are *$p < 0.05$, **$p < 0.001$, ****$p < 0.0001$ by ANOVA followed by Tukey's post hoc test. Flow cytometry gating strategy and additional datasets are shown in Supplementary Figs. 11, 12, 13. Source data are provided as a Source Data file.

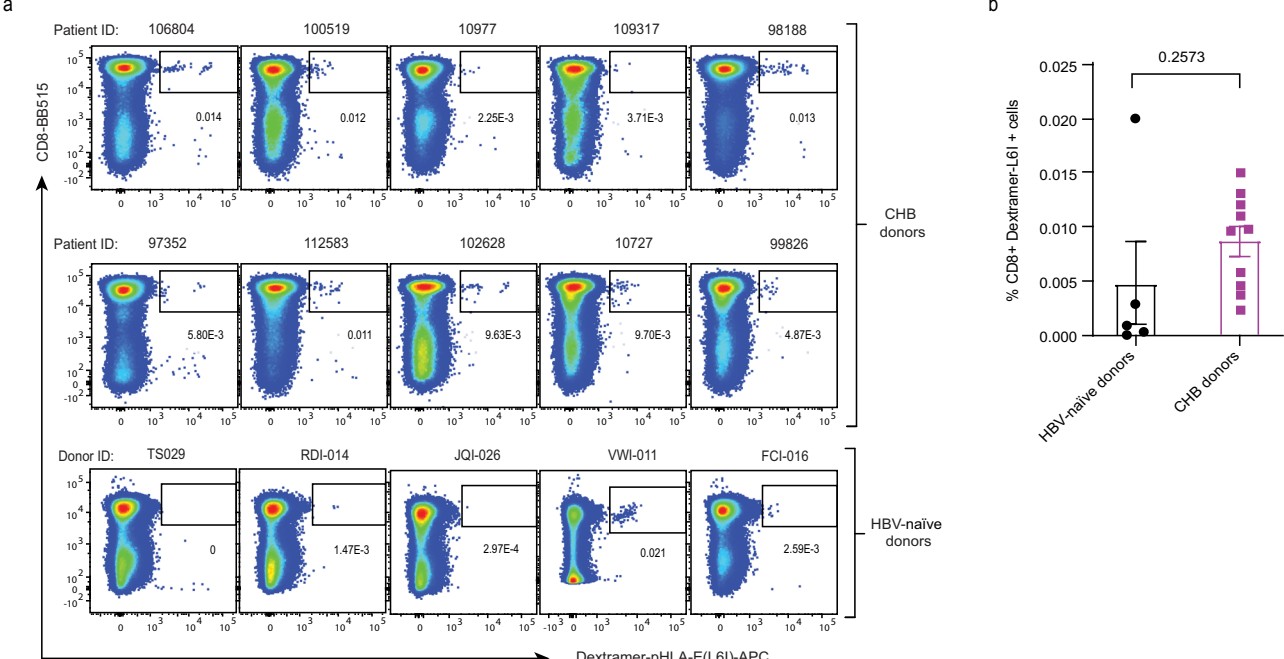

**Fig. 7 | Detection of HBV antigen-specific T cells by pHLA-E multimers. a** T cells from people with CHB (*n* = 10) and HBV-naïve donors (*n* = 5) were expanded in vitro by stimulation with streptavidin magnetic beads conjugated to Env$_{371-379}$ (L6I)-pHLA-E and anti-CD28 for two weeks and stained with dextramer-E and antibody cocktail. Dot plots show the percentage of CD8$^+$ dextramer-E$^+$ cells (dextramer

positive cells were gated as live CD3$^+$CD4$^-$CD8$^+$ singlet cells). **b** Quantification of % of CD8$^+$ dextramer-E positive cells. Statistical analyses were performed using two-tailed unpaired *t*-test in GraphPad Prism software (version 10.0.2), data represented as dot plot ± SEM (*n* = 5 for HBV-naïve donors (in black); *n* = 10 for people with CHB (in purple)). Source data are provided as a Source Data file.

encoding for either Env$_{371-379}$ (S3N) or Env$_{371-379}$ (L6I) peptides, consistent with higher affinity for HLA-E than the Env$_{371-379}$ peptide and consequently the ability to compete with canonical VL9 signal peptides for HLA-E binding[26].

As peptide levels produced in a minigene system may not be physiologically relevant, we tested the a09b08 ImmTAV molecule against the HCC cell line PLC/PRF/5, in which expression of the Env$_{371-379}$ (L6I) epitope is driven by transcription of integrated HBV DNA[41,44]. The killing of PLC/PRF/5 cells by ImmTAV-redirected T cells occurred with slow kinetics (over 27 h), which were accelerated (around 10 h) by overexpression of HLA-E*01:03 molecules. The slow kinetics were consistent with low endogenous HLA-E expression and almost undetectable numbers of pHLA-E Env$_{371-379}$ (L6I) molecules by mass spectrometry analysis. Endogenous antigen presentation was confirmed in a second model, using HepG2 cells transfected with replication-competent infectious 1.3-mer cDNA clones of HBV genotypes encoding for the three peptide variants. The a09b08 ImmTAV molecule redirected T cells only to targets expressing the Env$_{371-379}$ (L6I) peptide and treated with IFN-γ or overexpressing HLA-E molecules. We speculate that the lack of ImmTAV-dependent redirection to targets expressing the Env$_{371-379}$ (S3N) peptide could be due to a combination of lower affinity of this peptide for HLA-E molecules and limited peptide availability due to

transfection of only 10% of target cells. HLA-E-restricted a09b08 ImmTAV induced release of IFN-γ and GzmB from the effector T cells and inhibited viral replication. Although the strength of the response induced by HLA-E-restricted a09b08 ImmTAV was lower than that elicited by HBV HLA-A*02:01 ImmTAV, this can be explained by different density of peptide–HLA complexes for the two antigen-presenting molecules. Indeed, the activity of the two molecules was comparable when tested against HLA-E*01:03-transduced targets.

Consistent with previous studies[17], our findings reaffirm the ability of the a09b08 ImmTAV to redirect primary T cells and elicit a poly-functional immune response, including the release of IFN-γ and GzmB upon T cell activation. Together with TNF-α, IL-2, and IL-6[17], this cytokine profile may have indirect effects on suppressing viral replication through various mechanisms, such as inhibition of HBV entry, reduction of cccDNA levels, and decreased secretion of HBsAg[50,51].

Lastly, we detected HBV Env$_{371-379}$ (L6I) HLA-E-restricted CD8$^+$ T cells in the PBMC of people with chronic HBV and HBV-naïve individuals after in vitro expansion with Env$_{371-379}$ (L6I)-pHLA-E complexes. Although ex vivo identification of HLA-E-restricted CD8$^+$ T cells in people with chronic HBV proved elusive, a possible explanation could be the limited frequency of these cells in the blood, because of tissue residency. More importantly, HLA-E-restricted CD8$^+$ T cells were

detected at higher frequency in people with chronic HBV, providing evidence for the naturally occurring presentation of this epitope during infection.

The main limitation of this work is the lack of evaluation of the a09b08 ImmTAV molecule against HBV-infected primary human hepatocytes (PHH). HLA-E is expressed on the majority of primary hepatocytes and its expression is maintained on HBV-infected PHH[11]. We speculate that in the presence of an inflammatory hepatic microenvironment[52,53], HLA-E molecules could also be upregulated, promoting efficient recognition by an ImmTAV molecule.

Taken together, our findings demonstrate that the HBV Env$_{371-379}$ (L6I) peptide can compete with VL9 variants to bind HLA-E and be targeted with TCR-based immunotherapeutics when increased surface levels of HLA-E are present in cells of hepatic origin. Considering the distinct geographical distribution of HBV genotypes, we estimate that the CHB population coverage of the most stable Env$_{371-379}$ (L6I) variant worldwide would be around 29% (Supplementary Data 2).

To our knowledge, Env$_{371-379}$ is the first HBV peptide to be identified as an HLA-E epitope, adding to previous reports describing HLA-E restricted T cells targeting peptides from HIV-1[35,54], SARS COV2[34], and influenza[55], expanding our understanding of HLA-E biology. While the monomorphic nature of HLA-E offers the potential for a universal therapeutic, this study, together with our recent report describing the instability of the HLA-E peptidome in HIV[56], highlights the impact that peptide sequence variation may have on pHLA-E complex stability, potentially limiting the clinical utility of HLA-E targeting.

## Methods

### Computational prediction of HLA-E-binding peptides
HBV protein amino acid sequences from genotypes A, B, C, D, and E obtained from GenBank were aligned using MUSCLE and searched for 9-mer or 10-mer peptides within envelope proteins. The binding affinity of each peptide to HLA-E was predicted using netMHCpan4.0[57]. Seventy peptides were selected for assessment with stability assays (Supplementary Data 1).

### Computational prediction of peptide sequence conservation
To calculate the genotype prevalence of described HBV peptide variants, previously published per-genotype sequence nucleotide alignments were obtained from http://hvdr.bioinf.wits.ac.za/alignments (accessed 01/03/2019). Command line tool tblastn was used to search for the location of the peptide targets in each genotype alignment. Relevant nucleotide data was extracted from each alignment with complete coverage of the target, and then translated to amino acid sequences. Prevalence is presented as a proportion of the total number of samples analysed (Supplementary Table 1).

### Computational prediction of population coverage of peptide variants
Predicted population sizes of HBV-infected individuals by genotype, from ref. 58, were multiplied by the estimated proportion of Env$_{371-379}$ peptide variants per genome to obtain predicted population coverage. Per genotype estimates were summed (genotypes A–E) to estimate total targetable population size (Supplementary Data 2).

### Production of soluble pHLA-E complexes
Refolding and complex formation was done by dilution of $\beta_2$m, heavy chain and peptide into 200 mL of 100 mM Tris HCl, pH 8, 400 mM L-arginine HCl, 2 mM EDTA, 5 mM reduced glutathione, 0.5 mM oxidized glutathione, 0.5 mM phenylmethylsulfonyl fluoride. The final concentrations of the heavy chain, $\beta_2$m, and the peptide were 31 µg/mL (1 µM), 24 µg/mL (2 µM), and 10 µg/mL (10 µM), respectively. The refolding mixture was incubated at 10 °C for 24–36 h. The refold solution was passed through a 0.45 µM filter before loading the protein onto a POROS HQ column, washed with 1 M sodium hydroxide, water, and pre-equilibrated with 20 mM Tris, pH 8.1. Protein was eluted over an increasing gradient (0–50%) of 20 mM Tris pH 8.1 + 1 M NaCl. The eluate was collected in 1 mL fractions. Protein was concentrated to 1 mL by pooling all the fractions before loading onto a Superdex S75 gel filtration column washed with sodium hydroxide and equilibrated in PBS. Protein was eluted in 1 mL fractions, and samples were analysed by SDS-PAGE. For structural studies, non-biotinylated soluble HLA-E peptide complexes were produced using the HLA-E heavy chain without the AviTag™ sequence.

### pHLA-E stability assays
Thermal shift assays were performed using the Quantstudio 6 (Applied Biosystems; Waltham, MA). Peptides were obtained by chemical synthesis (Peptide Protein Research Ltd; Fareham, UK) and solubilized in DMSO. Peptides of interest were added to refolded and purified HIV Gag$_{275-283}$-HLA-E*01:03 complexes (in PBS at 0.25 mg/mL) at a 60:1 molar ratio, combined with SYPRO Orange protein gel stain (Thermo Fisher Scientific, Cat. No. S6651). This mixture was heated from 22 to 95 °C at 1 °C/min while detecting fluorescence using the FAM filter set, with excitation and emission wavelength at 495 and 518 nm, respectively. Positive hits (indicated by a typical melting curve) were analysed using Protein Thermal Shift Software version 1.4 (Thermo Fisher Scientific) to determine Tm. The stability of all pHLA-E complexes was assessed by SPR using a BIAcore T200 instrument (Cytiva, Marlborough, MA). Purified biotinylated pHLA-E monomers (HLA-E*01:03 was used throughout) were immobilized onto a streptavidin-coupled CM5 sensor chip. A total of 1 µM of soluble ILT2 was flowed over the chip at 10 µL/min for 60 s. ILT2 binding to pHLA-E complexes was measured at regular intervals, and responses were normalized by subtracting the bulk buffer response of a control flow cell containing no pHLA. Binding $t_{1/2}$ was calculated by plotting % activity against time using the Biacore T200 evaluation software version 3.0 and GraphPad Prism version 8.3.0. Cell surface stabilization of HLA-E on K562 cells were assessed as follows. K562 cells transduced with single chain HLA-E*01:03-β2m were either left unpulsed or pulsed with 10 µg/mL peptide for 2 h at 37 °C/5% $CO_2$. Immediately following peptide pulsing, cells were washed once with wash buffer (PBS + 2 mM EDTA + 2% human AB serum [Sigma-Aldrich, Cat. No. H3667]) and either left unstained or stained for 30 min at 4 °C using anti-human HLA-E-PE (3D12; BioLegend) or anti-mouse IgG1κ-PE (MOPC-21; BD Pharmingen) (Supplementary Table 9). Flow cytometry was performed using a Sony SH800S (Sony Biotechnology, software version 2.1.5.) loaded with a 100 µm sorting chip (Sony Biotechnology, Cat. No. LE-C3210) and calibrated with automatic setup beads (Sony Biotechnology, Cat. No. LE-B3001). Cytometer files were exported and analyzed with FlowJo software (FlowJo LLC version 10.7.1).

### Generation of TCR and ImmTAV molecule
A TCR specific to Env$_{371-379}$, Env$_{371-379}$ (S3N), and Env$_{371-379}$ (L6I) bound to HLA-E*01:03 was isolated from naïve TCR libraries. TCRs were selected by panning the phage libraries on immobilized pHLA-E. Briefly, the biotinylated pHLA-E complexes were captured in a Nunc immunotube coated with streptavidin (10 µg/mL in PBS). To isolate high-affinity TCRs, we decreased the concentration of biotinylated pHLA-E tenfold for each round of panning. The TCR phage were allowed to bind the immobilized pHLA-E for at least 2 h. Nonbinding phages were removed by sequential washing (10–20 washes of PBS + 0.1% Tween20 and 10 to 20 washes of PBS). Binding phage were then eluted from the immunotubes by adding 1 mL of 100 mM triethylamine, incubating for 10 min at room temperature (20–25 °C), transferring the solution to a new tube containing 0.3 mL of 1 M Tris-HCl, pH 7.0. Half of the eluted phage solution was used to infect 10 mL of *Escherichia coli* TG1 grown to OD$_{600}$ = 0.3–0.5 and supplemented with 5 mM Mg2+. After

incubation for 30 min in a water bath, bacteria were plated on TYE (10 g/L tryptone, 5 g/L yeast extract, 8 g/L NaCl, 15 g/L Bacto-Agar) plates containing 100 mg/mL ampicillin and 2% glucose and grown overnight at 30 °C. For negative selection HLA class Ia signal peptides pHLA-E complexes were used. Wild-type and affinity-enhanced TCR chains were fused to a CD3-specific scFv via a flexible linker to generate a bispecific retargeting molecule.

### Measurement of binding affinities and kinetics

Binding analysis of purified soluble ImmTAV molecule to pHLA-E complexes was carried out by surface plasmon resonance (SPR), using either BIAcore™ T200 (for weak affinity molecules) or a BIAcore™ 8 K system (for affinity-enhanced molecules). Briefly, biotinylated cognate peptide–HLA-E complexes were immobilized onto a streptavidin-coupled CM5 sensor chip. Flow cell one was loaded with free biotin alone to act as a control surface. $K_D$ values were calculated assuming Langmuir binding, and data were analyzed using a 1:1 binding model (GraphPad Prism [v8.3.0] GraphPad Software, San Diego, CA) for steady-state affinity analysis and Biacore Insight Evaluation [v2.0.15.12933] (Cytiva, Marlborough, CA) for single-cycle kinetics analysis. For the measurements using $Env_{371-379}$ (S3N) and $Env_{371-379}$ pHLAs 5 μM of peptide was added to the running buffer.

### Identification of human peptides for cross-reactivity screening

To minimize any downstream risk of off-target activity, we developed a bioinformatics pipeline to screen the entire human proteome (UniProt reference UP000005640) for any 9-mer peptides with some degree of homology to the HBV $Env_{371-379}$ target peptide. Homologous peptides were identified as being either (i) biochemically similar to the index peptide, and/or (ii) having fewer than four site-wise amino acid differences (Hamming distance <4) to the index peptide[59]. Biochemical similarity is defined by a "BLOSUM score", which was calculated as the mean of the position-wise BLOSUM62 substitution probability values between two peptides. Similar peptides were then screened for predicted HLA-E*01:01 binding by netMHCpan4.0 using a conservative binding threshold of <5%. The Genotype-Tissue Expression portal (GTEx; https://www.gtexportal.org/home) was then used to analyze the expression of genes containing the human peptides with some similarity to the target. Peptides were obtained by chemical synthesis (Peptide Protein Research Ltd) and solubilized in DMSO to a concentration of 4 mg/mL prior to use. See Supplementary Table 7 for peptide details.

### Flow cytometry to assess HLA-E surface expression

Cells were trypsinized, counted, and separated into $0.5 \times 10^6$ cells/sample. Cells were washed with wash buffer (PBS + 2 mM EDTA + 2% human AB serum (Sigma-Aldrich, Cat. No. H3667)) and either left unstained or stained at 4 °C for 30 min using Molecular Probes™ LIVE/DEAD™ Fixable Violet (Invitrogen Cat. No. L-34955) plus anti-mouse IgG1κ-PE (MOPC-21; BD Pharmingen) or anti-human HLA-E-PE (3D12; BioLegend) as per the manufacturer's instructions (Supplementary Table 9). Samples were washed twice before a minimum of 50,000 total events were analysed per sample using a Sony SH800S (Sony Biotechnology, software version 2.1.5.) calibrated with automatic setup beads (Sony Biotechnology, Cat. No. LE-B3001). Cytometer files were exported and analysed with FlowJo software (FlowJo LLC version 10.7.1) and GraphPad Prism v9.0.1.

### Flow cytometry with dextramer conjugated pHLA-E to stain Jurkat clones

Jurkat clones transduced with WT or high-affinity a09b08 TCRs were spun down, counted, and separated into $1.0 \times 10^6$ cells/sample. Cells were washed with wash buffer and stained with dextramer-APC (Klickmer-APC, Immudex, Cat no. DX01K, 01:10 dilution) conjugated to $Env_{371-379}$ (L6I) pHLA-E, $Env_{371-379}$ (L6I) pHLA-A*02:01 and Cw3 signal

peptide pHLA-E*01:03 for 30 min at 4 °C. Cells were then washed and stained with fixable viable dye (eFlour 450) and analysed on a BD LSR Fortessa machine.

### Protein crystallization

The TCR-pHLA complexes were prepared by mixing purified TCR and pHLA with excess peptide at a molar ratio of 1:1.5:1 and concentrating to approximately 10 mg/mL. The crystallization trials were set up by dispensing 150 nl of protein solution plus 150 nl of reservoir solution in sitting-drop vapor diffusion format in two-well MRC Crystallization plates using a Gryphon robot (Art Robbins). The plates were maintained at 20 °C in a Rock Imager 1000 (Formulatrix) storage system. Diffraction quality crystals of TCR a09b08 with each pHLA grew in 20% (w/v) PEG 3350, 100 mM BIS-TRIS propane pH 8.5, 200 mM sodium sulfate (Molecular Dimensions).

### X-ray data collection and structure determination

Crystals were cryoprotected using a reservoir solution supplemented with 30% (v/v) ethylene glycol and then flash-cooled in liquid nitrogen. X-ray diffraction data were collected at the Diamond Light Source (Oxfordshire, UK) on beamline I04. Diffraction images were indexed, integrated, scaled, and merged using dials[60] and dials.scale through the xia2[61] automated data-processing suite. Structures were solved by molecular replacement using PDB 5MEN as a model for TCR a09b08 and PDB 7NDQ as a model for HLA-E*01:03 and Beta-2-microglobulin in Phaser[62]. Models were built using iterative cycles of interactive model building in COOT[63] and refinement using Refmac5[64] in the CCP4 suite[65]. Additional model validation was performed using PDB_REDO[66]. The data processing and refinement statistics are listed in Supplementary Table 3. The structural figures were prepared using PyMOL (Schrödinger).

### Generation of minigene-expressing cell lines

HepG2 cells were transfected with a linearized minigene construct containing blasticidin selection marker and HBV peptides ($Env_{371-379}$ ILSPFLPLL, $Env_{371-379}$ (L6I) ILSPFIPLL or $Env_{371-379}$ (S3N) ILNPFLPLL) fused to ubiquitin. 24 h after transfection, cells were grown in cell culture media supplemented with 20 μg/mL blasticidin to select for minigene-stable cell clones.

### Affinity enrichment and mass spectrometry analysis of $Env_{371-379}$ (L6I)

PLC/PRF/5 wildtype and PLC/PRF/5 cells overexpressing HLA-E*01:03 were lysed in ice-cold lysis buffer (50 mM Tris pH 7.5, 150 mM NaCl, 0.5% NP-40) containing 1x HALT™ Protease and Phosphatase Inhibitor Cocktail (Thermo Scientific, 78447) by sonication for 30 s at 4 °C. Cell lysates were subsequently incubated for 1 h at 4 °C with agitation. Iodoacetamide (Sigma-Aldrich, I1149) was added to the lysate at a final concentration of 25 mM to alkylate cysteines during centrifugation at 15,000×g for 45 min at 4 °C. Five hundred μg biotinylated a09b08 TCR was bound to High Capacity Magne® Streptavidin (Promega, V7820) beads by incubation at 4 °C for 1 h. Biotinylated a09b08 TCR coupled streptavidin beads were subsequently incubated with cell lysates at 4 °C for 1 h using a rotating mixer. Beads were washed sequentially with wash buffer 1 (50 mM Tris-HCl pH 7.5, 150 mM NaCl, 5 mM EDTA, 0.05% NP-40), wash buffer 2 (50 mM Tris-HCl pH 7.5, 150 mM NaCl), wash buffer 3 (50 mM Tris-HCl pH 7.5, 450 mM NaCl), followed by wash buffer 2. Peptide–HLA complexes were eluted from the a09b08 TCR-bound streptavidin beads using 0.1% trifluoroacetic acid (TFA; Sigma-Aldrich, 302031), and peptides purified from eluates using C18 SepPak cartridges (Waters, WAT043395). Peptides were dried and stored at −80 °C before mass

spectrometry analysis. Lyophilized peptides were reconstituted in 5% acetonitrile (VWR, 83640.320) containing 0.1% TFA. Heavy stable isotope labeled internal standard peptides (JPT Peptide Technologies) were spiked into each sample before being analysed by mass spectrometry. All PRM data was analysed using Skyline[67] (version 21.1.0.278).

### PBMC and cell lines

Effector PBMC were isolated from whole blood obtained from anonymized healthy volunteers who consented to donate at Immunocore as part of a UK Health Research Authority-approved study. The study protocol (REC reference 13/SC/0226) was approved by the Oxford A Research Ethics Committee. Briefly, PBMC were isolated by density centrifugation using Ficoll-Hypaque. Cell lines were purchased and grown as specified in Supplementary Table 10. Cell line authentication and mycoplasma testing were routinely carried out by the LGC Standards Cell Line Authentication Service and Mycoplasma Experience Ltd, respectively. Several of these cell lines were modified as specified in Supplementary Table 10 and further described below. These cell lines cover a range of MHC Class I alleles and are, therefore, expected to express a variety of HLA signal peptides naturally presented by HLA-E (Supplementary Table 6).

### HLA-E genotyping

DNA was extracted from $0.5-3 \times 10^6$ cells using a QIAprep Spin mini kit (Qiagen). Genomic DNA was amplified using forward primer 5'-GGTCTCACACCCTGCAGTGGA-3' and reverse primer 5'-AGCCCTGTGGACCCTCTT-3'. DNA was PCR amplified with Phusion High Fidelity DNA polymerase (New England Biolabs) and migrated in 1.5% agarose gel. A band of ~280 bp was excised and purified using Nucleospin Gel and a PCR Clean-up kit (MACHEREY-NAGEL). Sanger sequencing was used to determine the polymorphism in codon 107 of HLA-E (Supplementary Table 11).

### Generation of THP-1 β2m and CIITA knockout cells

THP-1 cells were genetically modified using CRISPR-Cas9 nickase to eliminate endogenous B2M and CIITA proteins using a similar method to that described by ref. 68 sgRNA sequences used to target *B2M* were CTCGCGCTACTCTCTCTTTC (sense) and GGCCACGGAGC GAGACATCT (antisense); for targeting *CIITA*, sgRNA sequences were CTACCACTTCTATGACCAGA (sense) and CATCGCTGTTAAGAAGC TCC (antisense). *B2M/CIITA* double knockout cell clones were validated by targeted DNA sequencing (Supplementary Table 11).

### Generation of HLA-E overexpressing cell lines

To generate cell lines ectopically expressing HLA-E, plasmids were designed and cloned for use in lentiviral transductions. Human codon-optimized sequence of single chain dimer $\beta_2m$-HLA-E*01:01 and $\beta_2m$-HLA-E*01:03 were synthesized by GeneArt (Thermo Fisher Scientific) and cloned into the pELNS transfer vector using 5' NheI site and 3' SalI restriction sites. The full-length $\beta_2m$-HLA-E*01:03 constructs were kindly provided by Prof. Andrew Sewell (Cardiff University). To generate lentivirus, the plasmids described above were transfected into HEK293T cells using Turbofect™ transfection reagent (Thermo Fisher Scientific). Lentiviral particles were harvested and used to transduce cell lines, as indicated in Supplementary Table 10. Throughout the text, THP-1 *B2M and CIITA* knockout cells lentivirally transduced with $\beta_2m$-HLA-E*01:01 or $\beta_2m$-HLA-E*01:03 single-chain dimer are abbreviated to THP-1-E*01:01 or THP-1-E*01:03, respectively, and THP-1-E collectively.

### Enzyme-linked immunospot (ELISpot) assays

IFN-γ ELISpot assays were performed according to the manufacturer's recommendations (BD Biosciences). Briefly, target cells ($5 \times 10^4$ cells/well) were pulsed with a range of peptide concentrations for 2 h, washed once before resuspending in R10, and plating with ImmTAV molecule and PBMC (1:1 ratio). Plates were incubated overnight at 37 °C/5% $CO_2$, followed by IFN-γ detection, and spots were quantified using the BD ELISpot reader (Immunospot Series 5 Analyzer, Cellular Technology Ltd, Shaker Heights, OH, USA).

### Antibodies for blocking assays

For ELISpot assays including monoclonal antibodies, endotoxin- and azide-free unconjugated monoclonal antibodies anti-HLA-E (3D12) and anti-HLA-A2 (BB7.2) were generated to order (InVivo BioTech Services, Germany). Antibodies were added to target cells prior to addition of ImmTAV and PBMC, with a final antibody concentration of 10 μg/mL for the duration of the coculture (Supplementary Table 9).

### IncuCyte killing assay

The IncuCyte S3 live-cell analysis system (Essen Bioscience, Newark, UK) was used to perform killing assays with $Env_{371-379}$ (L6I)-expressing ($Ag^+$) and non-expressing ($Ag^-$) HCC target cells and PBMC from HBV-naïve donors. Briefly, target cells were stained with CellTracker Deep Red Dye (Invitrogen, Carlsbad, CA, USA). PBMC were added at a 10:1 ratio to targets with increasing concentrations of ImmTAV. IncuCyte Caspase-3/7 Green Apoptosis Assay Reagent (Essen Bioscience) was added to all wells, and plates were incubated at 37 °C/5% $CO_2$ with images taken every 3 h. The number of apoptotic events/$mm^2$ was calculated from two-color images. The analysis mask included size and eccentricity filters to exclude effector cells from the analysis.

### Chronic hepatitis B cohort

Ten people with CHB were recruited by Sanguine Biosciences®. PBMC and serum samples were obtained. The inclusion criteria for this study were: age 18–85 years; confirmed diagnosis of CHB based on detectable HBV DNA levels in the blood at the most recent clinic visit and/or no antiviral therapy for at least three months. Exclusion criteria were: receipt of any investigational product within 30 days of sample collection; concurrent infection with other hepatitis viruses or HIV. Ethical approval for the study was obtained by Sanguine, and all participants gave written informed consent.

### Viral DNA isolation, amplification, and sequence analysis

Viral DNA from the HBV patient serum was extracted from ~700 μL of serum using a QIAamp Blood mini kit according to the manufacturer's protocol. After extraction of DNA the HBV surface antigen region was amplified using universal PCR primers for the surface antigen[69]. Primers used in this study: forward - 5' GACTYGTGGTGGACTTCTC 3'; reverse – 5'-TCAGCAAAYACTYGGCA-3' and further amplified by nested PCR primers: forward - 5'-TGGATGTGTCTGCGGCGTTTTATCAT-3'; reverse 5'-ATDCKTTGACANACTTTCCAATCAA-3'[70]. PCR products were isolated and sent for sequencing by Sanger's sequencing using the primer: 5'−CACHTGTATTCCCATCCCA−3' to identify the HBV $Env_{371-379}$ peptide variant present in the HBV DNA sample (Supplementary Table 10).

### HBV infection model

HepG2 wildtype and HLA-E*01:01 and HLA-E*01:03 overexpressing cells were transfected with plasmids[45] encoding HBV-genotype A2 ($Env_{371-379}$ (L6I)), C1 ($Env_{371-379}$ (S3N)), and C2 ($Env_{371-379}$) using FuGENE6 transfection reagent (Promega) according to the manufacturer's instructions. The 1.3-mer cDNA clones of HBV-genotype constructs were kindly provided by Prof. Peter Revill (Melbourne Health). Transfection efficiency was assessed by intracellular staining with FITC anti-HBsAg antibody (Abcam). Twenty-four hours post-transfection, wildtype HepG2 cells were stimulated with 1 ng/mL of IFN-γ overnight to upregulate HLA-E expression, washed and

cocultured with effector (pan T cells from three HBV-naïve donors) with or without a09b08 ImmTAV, HBV HLA-A*02:01 ImmTAV[17] or an irrelevant Mtb RLPA HLA-E ImmTAB[46], at concentration of 1 and 10 nM. Some T cells were harvested at 72 h, stained with a cocktail of antibodies (CD2-PerCP Cy5.5 clone RPA-2.10, CD4-APC/Cy7 clone SK3, CD8 BV711 clone RPA-T8, CD69-APC clone FN50, and CD25-PE clone M-A251 from Biolegend) and analysed by flow cytometry on BD LSR Fortessa X-20. Culture supernatants were harvested on day 4, and fresh media was added to the cells for a further 48 h coculture. Culture supernatants were harvested again on day 6. IFNγ and GzmB in culture supernatants at day 4 post coculture were quantified using an MSD's U-PLEX custom assay (Meso Scale Technologies, LLC) as per the manufacturer's instructions. Secretion of HBeAg and HBsAg in the culture supernatant at day 4 and day 6 post coculture was quantified using CLIA kits (Ig Biotechnology LLC, USA) following the manufacturer's recommendations.

### Analysis of HBV Env$_{371-379}$ (L6I)-pHLA-E-restricted CD8$^+$ T cells
PBMC from people with CHB were obtained from Sanguine Biosciences®. Streptavidin-coated magnetic beads conjugated to refolded Env$_{371-379}$ (L6I)-pHLA-E and anti-CD28 antibody were used as artificial antigen-presenting cells (aAPCs) for the expansion of Env$_{371-379}$ (L6I)-pHLA-E-restricted T cells. Pan T cells (purified using the Miltenyi isolation kit) were stimulated with aAPCs twice over 2 weeks, with E:T of 2:1, and cultured for 1 week, then T cells were stimulated again with aAPCs and cultured for one more week in medium supplemented with 40 U/mL IL-2, 1700 U/mL IL-7, and 20 U/mL IL-15. Cells were subsequently analysed by flow cytometry with Env$_{371-379}$ (L6I)-pHLA-E dextramers-APC for 30 min at 4 °C. Cells were then washed and stained with a cocktail of antibodies (CD3-APC-Fire 750 (SK7), CD4-PerCP/Cy5.5 (RPA-T4), CD8-BB515 (RPA-T8), fixable viable dye (eFlour 450) before fixing and analysis on a BD LSR Fortessa machine (Supplementary Table 11). The gating strategy is described in (Supplementary Fig. 15).

### Reporting summary
Further information on research design is available in the Nature Portfolio Reporting Summary linked to this article.

## Data availability
The crystallography data generated in this study have been deposited in the RCSB protein data bank (PDB) with the accession codes 8RLT, 8RLU, and 8RLV. All data are included in the Supplementary Information or available from the authors, as are unique reagents used in this Article. The raw numbers for charts and graphs are available in the Source Data file whenever possible. Source data are provided with this paper.

## Code availability
Scripts for data processing and analysis are publicly available at https://github.com/Immunocore/FindPeptideOrthologs.

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

## Acknowledgements

The authors would like to thank Andrew Sewell (Cardiff University) and Peter Revill (Melbourne Health) for providing reagents; Daniel Fonseca for figure generation; Kate Vowell for sourcing patient sample material; Hussein Al-Mossawi and JoAnn Suzich for critical review of the manuscript; Giovanna Bossi, Lorraine Whaley, Nicola Smith, and Marco Lepore for scientific input.

## Author contributions

G.M., R.L.P., R.K., V.I., R.J.S., M.M.C., V.K., R.P., A.J., J.D., T.H., W.B., G.P., K.O., D.K., A.S., C.B., R.R., C.P., T.G., M.D., D.G., M.H., and L.F.G. performed the experimental studies. G.M., R.L.P., R.K., R.J.S., M.M.C., V.K., R.P., A.J., J.D., T.H., W.B., K.O., R.R., M.D., M.H., R.J.C., K.E.A., L.D., A.K., S.L., M.S., and L.F.G. carried out the analysis. A.T. and A.W. performed the computational studies. M.S. and L.F.G. supervised the work. G.M., R.L.P., R.K., V.I., V.K., R.P., T.H., L.D., M.S., and L.F.G. wrote the manuscript. All authors commented on the manuscript.

## Competing interests

G.M., R.L.P., R.K., V.I., R.J.S., M.M.C., V.K., R.P., A.J., J.D., T.H., W.B., G.P., K.O., D.K., A.S., C.B., R.R., C.P., T.G., A.T., A.W., M.D., D.G., M.H., R.J.C., K.E.A., L.D., A.K., S.L., M.S., and L.F.G. are or were employees of Immunocore Ltd.
