## [Transparent Peer Review file · Nature Communications]

Viral sequence determines HLA-E-restricted T cell recognition of hepatitis B surface antigen

Corresponding Author: Dr Luis Godinho

Version 0:

Reviewer comments:

Reviewer #1

(Remarks to the Author)

This manuscript describes the development of a soluble affinity-enhanced TCR anti-CD3 bispecific molecule able to recognize HBV envelope peptides presented by HLA-E in cell lines transfected with HBV envelope or naturally producing HBV envelope derived from natural HBV-DNA integration. The authors demonstrated that only some variant ENV peptides are able to stably bind this bi-specific molecule and redirect T cells against the ENV-expressed targets. They also show that some HLA-E restricted T cells are present, even if at very low frequencies, in CHB patients suggesting the existence of HLA-E restricted HBV specific T cells in HBV infection.

The work is robust and novel. Data support the conclusions made. The importance and limitations of their findings are discussed in a very balanced way.

This reviewer's expertise is more on HBV pathogenesis and T cell response, and as such, I cannot fully judge the initial characterization of HBV peptides binding to HLA-E (figure 1-2) and of the structural modification of the TCR (figure). The experiments performed to test the functional ability of the bispecific molecule to activate T cells against target pulsed with HBV peptides or expressing HBV envelope antigen are convincing and performed with appropriate controls.

I have only a single major comment that perhaps the authors can address in their discussion.

Overall, the study lacks an analysis of whether HBV-infected hepatocytes might be a good target for such a bispecific molecule. The authors only test their bispecific molecule on targets that were HBV transfected but not infected by HBV. (This distinction is clear in the text of the paper. However, in the figure legends, the authors use the term "HBV-infected Hep G2," which is not correct. Please modify). In addition the cell lines used are transformed hepatoma cells and not normal hepatocytes. This review fully understands that experiments of HBV infection of normal hepatocytes are complex (particularly in the UK, where HBV infection, I think, can be handled only in Cat 3 lab), and performed an overexpression of HLA-E in normal hepatocytes might not be feasible. However this limitation should be clearly pointed out in their discussion. I am also wondering whether HLA-E is also expressed at low level in normal hepatocytes (authors wrote in the text that Hep G2 have low HLA-E expression, but what about normal hepatocytes?) and whether data are available about the impact of HBV infection on the expression of HLA-E. Such points might be expanded in the discussion.

Minor

1) HBV-specific CD8 T cells are notoriously present at low frequency in CHB patients but are usually more easily detectable (especially after in vitro expansion) in patients who resolved HBV infection. Why the authors didn't try to test the presence of HLA-E-restricted HBV-specific T cells in patients who resolved HBV infection? This patient population can be a very nice addition of the paper. Authors should also specify whether healthy individuals tested as controls were vaccinated with HBV vaccine.

2) Table S1 can the authors add the sequence AA numbers of the peptides?

3) Table 7 Are all the patients HBsAg+? Not clear.

Reviewer #2

(Remarks to the Author)

In the manuscript 'Viral sequence determines HLA-E-restricted T cell recognition of hepatitis B surface antigen', Paterson et al identify a novel HLA-E restricted epitope from HBV, characterize its function, and test its ability to be targeted both by endogenous T cells and a TCR/anti-CD3 protein engager developed by Immunocore. This work is an interesting

contribution to the expanding field of non-canonical HLA-E bound peptides, and underlies some of the promise and challenges of exploiting these as therapeutic targets. I am enthusiastic about the work, but it would benefit from some additional rigor and validation before publication:

Major points

1. Since there is some evidence suggesting HLA-E can be refolded in the absence of peptide and remain B2M-associated (such as in Walters et al Nature Comms 2018), it would therefore be helpful to establish a baseline level of ILT2 binding (or demonstrate lack of binding) for peptide-free HLA-E and subtract this from the peptide-associated HLA-E measurements.
2. All plots in Figure 1 lack error bars reflecting SD between 3 independent biological repeats.
3. Could the authors comment on why the relatively minor L6I change results in such a large increase in cell surface stabilization? For example, is there a structural basis for this dramatic increase in stability and cell surface expression? It's curious that a single mutation from one medium sized hydrophobic residue to another medium sized hydrophobic residue at the secondary anchor position 6 could result in such a dramatic increase in cell surface stabilization resulting to substantially higher levels relative to canonical VL9 variants. It would be helpful to have these measurements (which can often be noisy) conducted in triplicate and shown with error bars, and to have the L6I variant shown as a thermal melt.
4. In the co-complex structure, the HBV-derived Env371-379 S3N variant makes an additional hydrogen bond with Gln-156 on the alpha 2 helix of HLA-E. The authors suggest this could be the structural basis for higher HLA-E cell surface stabilization levels relative to wild type Env371-379 which lacks this hydrogen bond. The L6I variant also lacks this additional hydrogen bond yet stabilizes both HLA-E*01:01 and HLA-E*01:03 at higher levels than other Env371-379 mutants including the wild-type variant. This L6I mutant also stabilizes cell surface HLA-E at higher levels than canonical VL9 peptide. Can the authors comment on the superior stabilization by the L6I variant in relation to the structural data? Are there any structural features unique to the L6I structure which would correlate with higher peptide-HLA-E complex stability? The chosen affinity enhanced TCR contains an N96 TCR beta residue which hydrogen bonds to the backbone of the peptide at position 6. Interactions between TCR and the peptide backbone can be independent of peptide side chains. As the HBV Env-derived peptides adopt a similar peptide backbone conformation to VL9, this additional hydrogen bond to the P6 peptide backbone could feasibly also occur for VL9 peptides. Given the negative consequences for a cross-reactive binder, it would be helpful to experimentally validate lack of HLA-E/VL9 binding by a09b08 TCR via SPR.
5. For the functional assays, it is difficult to assess how the amount of output in the ELISPOT or cytokine release assays compares to (for example) targeting an HLA-A2-restricted response. A direct comparison with a previously described ImmTAV would be helpful calibration.
6. The viral inhibition results in Figure 6G-H show what appear to be an extremely modest effect. Can the authors provide some type of comparison with another ImmTAV? And could a negative control be included? I would also consider moderating language about these results given the largest outcomes seem like ~2-4 fold differences.
7. Given that HBV Env371-379 L6I was the only variant tested which resulted in T cell activation, cytokine secretion and antiviral activity in the viral infection model (albeit in the presence of IFN- γ pre-treatment), could the authors elaborate further on the prevalence of this particular variant in the discussion? For example, what would the associated therapeutic coverage among individuals infected with HBV look like?
8. HLA-E/VL9 specific CD8+ T cells have been reported to be present at high frequencies, averaging 10% of total CD8+ T cells in CMV+ donors (Sullivan et al Sci Immunology 2021). Therefore, it would be nice if the authors included HLA-E/VL9 dextramers as negative controls in their in vitro priming and subsequent staining experiments to rule out cross-reactivity. Dextramer staining to assess potential TCR cross-reactivity between HLA-E vs HLA-A2 presenting the HBV Env variants would also constitute important controls here as this peptide binds to HLA-A2, and could explain why HBV+ donors had higher dextramer staining relative to healthy donors if prior expansion was in fact being driven by HLA-A2.
9. In the discussion the authors mention that mass spec analyses were conducted to identify HLA-E presentation of HBV Env peptide variants but that even the highest affinity L6I variant was almost undetectable via this approach. Would the authors consider including these data in the supplementary? For example, it would be helpful to the field to know whether the authors used peptide pulsing versus endogenous peptide presentation methods and whether highly sensitive SIL approaches were attempted?
10. In the discussion, the section: 'Taken together, our findings demonstrate that the HBV Env371-379 (L6I) peptide can compete with VL9 variants to bind HLA-E and be targeted with TCR-based immunotherapeutics.' should be softened or followed with the caveat that such immunotherapeutic targeting would likely only be effective in combination with IFN- γ treatment or in a setting where HLA-E surface expression on HBV-infected cells was elevated.

Minor points

1. The FACS histograms in Supplementary Figure 1 which depict fluorescence intensity are more informative in terms of HLA-E expression density than the plots in Figure 1B which simply portray the % of cells in the HLA-E+ gate. The authors might want to consider switching these plots between the main and supplementary figure packets.
2. The wording of the sentence between lines 174-177 is slightly unclear.: 'The peptide contacts are predominantly through hydrophobic interactions, including the TCR alpha chain H94 stacking with I1 and P4...'

It sounds as though the authors are stating a list of hydrophobic interactions between the TCR and peptide. However, the final contact mentioned in this list is between the TCR beta chain Arg 95 and peptide Leu 8 which appears to be a hydrogen bond, not a hydrophobic interaction?

3. It would be helpful if the authors could report the CMV seropositivity and HLA-A2 positivity rates in the healthy and HBV PBMC donors used in these experiments.

4. Table S1 should be reformatted in some way to be readable with all columns on the same page.

5. Table S2 appears to have a duplication for structure 8RLV (the I/sigI term looks like it was copied over by the Rmerge)

Reviewer #3

(Remarks to the Author)

In this manuscript, Paterson et al. identified an HLA-E restricted HBV peptide antigen derived from the HBV envelop (Env) protein (named Env371-379) as well as its two peptide variants (S3N and L6I) and designed a bispecific ImmTAV molecule to redirect T cell recognition towards these targets. The ImmTAV molecule is composed of an Env371-379/HLA-E specific, affinity-enhanced T cell receptor (TCR) a09b08 engineered by the authors and an anti-CD3 single chain antibody. This molecule is supposed to bridge functional T cells together with the Env371-379/HLA-E-positive target cells and can thus bypass the antigen specificity of individual T cells to the target. A series of biochemical, structural, and functional analysis have been done on either the TCR or the ImmTAV molecule to prove its potential in directing specific T cell response against the Env371-379/HLA-E antigens. By targeting a highly conserved HLA-E epitope, the overall idea of this work provides an interesting strategy that can lead to development of a relatively universal treatment to chronic HBV infections comparing to current solutions. However, in terms of actual utility of this strategy, I do have some concerns hoping to be addressed by the authors.

Major questions:

1. Regarding the presentation of the Env371-379/HLA-E target and relative stabilities:

From the perspective of stability, the authors showed in both Table S1 and Figure 1A that the thermal stability of the original Env371-379 peptide in complex with HLA-E was around 47 °C. A measurement by SPR showed a $t_{1/2}$ of 6.7 min for this peptide. How does this compare with other naturally presented HLA-E peptides, such as the signal peptide? Although the SPR measurement showed the S3N and especially the L6I variants have longer half-lives than the original Env371-379, are there T_m values for them as well – if not, these should be included? For the SPR stability measurement using ILT2 protein, is there any control measurement testing the stability of either Mtb RLPA or signal peptides in complex with HLA-E as a reference? This is important for the points below.

Following these questions, in presentation by HLA-E, the selected HBV peptide needs to compete with the existing signal peptides in HLA-E presentation. The relative binding values are important here, as is the comment in the discussion about detection via mass spectrometry. Have the authors pulsed the HLA-E-positive cells with a mixture of HBV and VL9 peptides at different ratios in the function analyses? Overall, the authors need to perform a stronger job at ascertaining the target and potential competitors/cross-reactive epitopes (see cross-reactivity point below).

The authors also mention that the Env371-379 peptide can be presented by HLA-A*02:01 allele as well. What is the stability of the various peptides in complex with HLA-A*02:01? Is there a possibility that the Env371-379 L6I peptide is preferentially presented by HLA-A*02:01, and could this be a limiting factor in targeting?

2. Regarding cross-reactivity:

Although the authors have put in considerable efforts for minimizing the potential cross-reactivity of a09b08TCR/ImmTAV towards other HLA-E presented other peptides, is there a possibility that the a09b08TCR can cross-react with other peptide/HLA-E antigens with an affinity weaker than that for the Env peptides/HLA-E but still above the threshold to activate T cells, especially with high levels of antigen presentation? For example, in Table S5, although the INF-g release for other peptides are quite low compared to the Env371-379 (L6I), there is still some signal with 1 nM usage of ImmTAV.

This is particularly relevant given 1) the questions regarding HLA-E/peptide stability above, 2) the statement on lines 168-171 that “the three HBV Env peptides adopt a similar binding conformation to the previously described binding of Mtb55 and canonical VL9 signal peptides to HLA-E”, and 3) the TCR’s picomolar affinity to the HLA-E/Env peptides. If there is really no detectable cross-reactivity to those epitopes, the authors should address why, or present their conclusions with more caution. Clearly, the potential for cross-reactivity is crucial for any therapeutic development and this should be addressed very carefully.

3. On the structures:

In Table S2, the I/sigma for the outer shell is different from those values in the validation reports and are all quite low (~0.5). Can the authors double check those numbers? Also, if the numbers in Table S2 are correct, does the highest resolution shell with I/sigma around 0.5 really help with the improvement of structure refinement, or is this essentially noise? The value of 0.5 is well below what is normally used as a cutoff.

In lines 166-168, the authors gave some explanation for why the S3N variant is more stably presented by HLA-E than the Env371-379. However, according to the stability measurement by SPR, the L6I stability ($t_{1/2} \sim 2$ h) increased the most from the Env371-379 ($t_{1/2} \sim 6.7$ min) comparing to the S3N variant ($t_{1/2} \sim 39.5$ min). Is there any structural reasoning for the higher stability of L6I?

For the statement on lines 168-171, can the authors support it with an alignment of the Env peptides/HLA-E and Mtb55 and canonical VL9 signal peptides/HLA-E structures?

In Figure 3A, the authors should show the peptide density map to support the peptide structural comparison. This should be an omit map to avoid bias, and at a size to clearly assess density and fit.

In Figure 3C, some amino acid labels are incorrectly indicated: Env371-379 structure: L5; Env371-379 L6I structure: L6.

4. Other receptor reactivity:

Since HLA-E is a ligand for NK cells, does the ImmTAV influence NK cell function via any NK receptor (CD94/NKG2A, B, C, etc.) interactions with HLA-E?

Minor comments:

a. Line 81: The usage of term "HBeAg" is confusing without any explanations for what it stands for.

b. Lines 117-119: the statement, "this peptide is presented in all forms of envelope proteins (large, medium, and small)" is misleading (overemphasized) without mentioning its prevalence in different genotypes. Can the authors provide accession numbers for some of the sequences used as an example to support the statement?

c. Line 162: The authors should indicate somewhere in the main text about which HLA-E allele was solved in crystal structures. The only position I could clearly find an answer to this is in the PDB validation report.

d. Lines 179-180: I think the statement from the authors that "the interactions between the a09b08 TCR and the HLA-E heavy chain are the same with all three peptides" is a little casual, as Fig.3D indicates residues of the TCR only interacting with one of the three peptide/HLA-E targets. Even though the contribution of these interactions can be limited, they're still considered to be different unless the authors can provide an explanation for why they're not significant.

e. Figure 2C: The unit for standard deviations looks to be M in this format.

f. Figure 3A: It's quite untraditional to see pMHC in a top-down view as shown. I understand that the authors may have their own considerations on this strategy, but this is difficult to interpret. If the point is to show the similarity of the three peptide structures, authors should show a structural alignment of all peptides without the distraction from the HLA-E binding groove.

g. Figure S2A: It could be due to the low resolution of figure, but I could not see error bars in all the figures.

Reviewer #4

(Remarks to the Author)

Reviewer #5

(Remarks to the Author)

Version 1:

Reviewer comments:

Reviewer #1

(Remarks to the Author)

The authors modified their statements and answered my question. It remains a little disappointing that they cannot detect higher frequencies of HLA-E restricted T cells in acute HBV patients, but the data indicate, if not an expansion of T cells specific for HLA-E/HBV peptide at least the ability of such HLA-class I to present HBV antigen.

Reviewer #2

(Remarks to the Author)

I thank the authors for their revisions, which strengthen the science. No more comments from me, I endorse publication.

Reviewer #3

(Remarks to the Author)

The authors have addressed all of my prior concerns in their revisions.

Reviewer #4

(Remarks to the Author)

Reviewer #5

(Remarks to the Author)

Reviewer 1

We thank this reviewer for acknowledging that the work is robust and novel and that we have discussed in a balanced way the importance and limitations of our findings.

The authors tested their bispecific molecule on HBV transfected targets but not infected by HBV. As such the study lacks an analysis of whether HBV infected hepatocytes might be a good target for such bispecific molecule. The distinction between infection and transfection is clear in the text of the paper. However, in the figure legends the authors use the term “HBV-infected Hep G2” that is incorrect. Please modify.

Thank you for picking up on the error in the figure legend, we have modified the figure 6 legend, and it now reads “HBV-transfected HepG2”.

In addition, the cell lines used are transformed hepatoma cells and not normal hepatocytes. This review fully understand that experiments of HBV infection of normal hepatocytes are complex (particularly in UK where HBV infection I think can be handle only in Cat 3 lab) and also performing an overexpression of HLA-E in normal hepatocytes might not be feasible. However, this limitation should be clearly pointed out in their discussion. I am also wondering whether HLA-E is expressed at low level in normal hepatocytes (authors wrote in the text that Hep G2 have low HLA-E expression, but what about normal hepatocytes? And whether data are available about the impact of HBV infection on the expression of HLA-E on hepatocytes. Such points might be expanded in the discussion.

As acknowledged by this reviewer, HBV infection of normal hepatocytes with or without additional HLA-E transduction is challenging. We have added this to our discussion in the form of limitations of our study. We also refer to the work of Burwitz et al., 2020 (PMID: 32161099) who detected low HLA-E surface expression on 90% of human donor primary hepatocytes (PH) ex vivo (Fig. 3B of the above reference), which was maintained after HBV infection (Fig. 3D and E of the above reference). As expected, HLA-E surface expression was lower than HLA class I.

Additionally, a higher level of HLA-E expression has been observed in hepatocellular cell carcinoma (HCC) tissues compared to adjacent non-cancerous liver tissues and to normal liver (PMID: 28197391, PMID: 31602270, PMID: 21744989).

HBV-specific CD8 T cells are notoriously present at low frequency in CHB patients and often not able to expand in vitro. HBV-specific CD8 T cell are usually more easily detectable (especially after in vitro expansion) in patients who resolved HBV infection. This could explain the low growth of HLA-E restricted HBV specific T cells in CHB patients showed in figure 7. However, why did the authors not test the presence of HLA-E-restricted HBV-specific T cells in some patients who resolved HBV infection? The presence of HLA-E-restricted T cells in some resolved patient population can be a very nice addition to the paper since the data of figure 7 (frequency of HLA-E-restricted HBV-specific T cells in CHB patients) are not fully convincing. The authors should also specify whether healthy individuals tested as controls were vaccinated with HBV.

We thank the reviewer for the suggestions. We obtained PBMC from people with CHB from a commercial source, which provided the following clinical data, included in supplementary table 8: confirmed chronic HBV, detectable HBV DNA and HBsAg levels in the blood and/or off HBV

treatment for 3 months or more. Donors 97352, 10977 and 112583 have serological profile consistent with resolved infection since they had undetectable blood levels of HBsAg at the time of sampling, with the caveat that this is based on a single time-point. However, in these donors the percentages of CD8+ dextramer L6I + cells were comparable to the other samples (0.06, 0.01 and 0.011%, respectively).

We have no access to the HBV vaccination status of the healthy individuals tested as controls. However, we do not believe this would alter our conclusions, as the licensed subunit vaccine induces strong antibody and CD4 responses, while HBsAg reactive CD8 T cells are rarely detected (PMID: 33898628).

Minor points

Table S1 can the authors add the sequence AA numbers of the peptides?

We have added the AA numbers of the peptides listed in the supplementary Data 1. Peptide amino acid numbering was retrieved from a full-length HBV surface antigen protein (Uniprot: P17398).

Table 7 Are all the patients HBsAg+? Not clear.

Seven of the 10 patients studied were HBsAg-positive at the time of sampling, while 3 had HBsAg levels below the limit of detection. We have modified supplementary table 8 to include the HBsAg status.

Reviewer 2

We thank this reviewer for acknowledging that this work is an interesting contribution to the HLA-E field and highlights promises and challenges of exploiting this axis therapeutically.

Major points

Since there is some evidence suggesting HLA-E can be refolded in the absence of peptide and remain B2M-associated (such as in Walters et al Nature Comms 2018), it would therefore be helpful to establish a baseline level of ILT2 binding (or demonstrate lack of binding) for peptide-free HLA-E and subtract this from the peptide-associated HLA-E measurements.

We thank the reviewer for highlighting Walter's work suggesting that HLA-E can be refolded in the absence of peptide, forming species which can be resolved as diffuse forms in blue native polyacrylamide gel electrophoresis. We have also refolded HLA-E*01:03 in the absence of peptide, however, as shown below and in agreement with Walter's results, we could not obtain a T_m value as a measure of thermostability. We also could not accurately measure the stability (t_{1/2}) of peptide-free HLA-E*01:03 molecules, since after the first ILT2 injection the percentage of active complex was below 20%.

Differential scanning fluorimetry (DSF) of empty refolded HLA*01:03 and HLA*01:03 Cw3 leader complexes. Melting curves (top panel) and melting peaks (bottom panel); pHLA-E complexes melting temperature (T_m) are shown for Cw3 leader complexes.

All plots in Figure 1 lack error bars reflecting SD between 3 independent biological repeats.

The thermal shift assay to screen the 70 peptides identified by bioinformatic approach (Figure 1a) was done with technical, not biological replicates. The error bars (SD) were included in all the figures, however, in some graphs, the standard deviation is so small that the bars are hardly seen.

Could the authors comment on why the relatively minor L6I change results in such a large increase in cell surface stabilization? For example, is there a structural basis for this dramatic increase in stability and cell surface expression? It's curious that a single mutation from one medium sized hydrophobic residue to another medium sized hydrophobic residue at the secondary anchor position 6 could result in such a dramatic increase in cell surface stabilization resulting to substantially higher levels relative to canonical VL9 variants.

We agree with the reviewer that it is interesting that mutation L6I leads to a marked increase in cell surface stabilization relative to the index sequence and the variant S3N. As discussed in the text, the structural analysis did not provide a clear explanation for these differences. It is however possible that the L6I peptide might be taken up and loaded more efficiently than the VL9 variants, as shown in the FACS plots in Figure 1b and supplementary figure 1.

It would be helpful to have these measurements (which can often be noisy) conducted in triplicate and shown with error bars, and to have the L6I variant shown as a thermal melt.

We have performed these experiments in triplicate and Fig. 1c now shows the $t_{1/2}$ and T_m values of the three refolded HLA-E*01:03 HBV Env₃₇₁₋₃₇₉ peptide variant complexes. We have also included values for HLA-E*01:03 complexes with two signal peptides from leader regions of Cw3 and A1 molecules for comparison.

In the co-complex structure, the HBV-derived Env371-379 S3N variant makes an additional hydrogen bond with Gln-156 on the alpha 2 helix of HLA-E. The authors suggest this could be the structural basis for higher HLA-E cell surface stabilization

levels relative to wild type Env371-379 which lacks this hydrogen bond. The L6I variant also lacks this additional hydrogen bond yet stabilizes both HLA-E*01:01 and HLA-E*01:03 at higher levels than other Env371-379 mutants including the wild-type variant. This L6I mutant also stabilizes cell surface HLA-E at higher levels than canonical VL9 peptide. Can the authors comment on the superior stabilization by the L6I variant in relation to the structural data? Are there any structural features unique to the L6I structure which would correlate with higher peptide-HLA-E complex stability?

Please see the previous comment regarding minor structural differences observed in the L6I variant. We have included a supplementary figure 5 comparing S3N and L6I variants with the index peptide surrounding peptide positions 3 and 6, respectively.

The chosen affinity enhanced TCR contains an N96 TCR beta residue which hydrogen bonds to the backbone of the peptide at position 6. Interactions between TCR and the peptide backbone can be independent of peptide side chains. As the HBV Env-derived peptides adopt a similar peptide backbone conformation to VL9, this additional hydrogen bond to the P6 peptide backbone could feasibly also occur for VL9 peptides. Given the negative consequences for a cross-reactive binder, it would be helpful to experimentally validate lack of HLA-E/VL9 binding by a09b08 TCR via SPR.

We had considered a potential cross-reactivity of the a09b08 TCR to HLA-E/VL9 complexes via interactions between the N96 TCR beta residue and the backbone of the peptide at position 6. We have tested the binding affinity of the a09b08 ImmTAV against a panel of pHLA-E/VL9 complexes and no binding was observed at 8.5 nM concentration, which is at least 10-fold higher than the predicted therapeutically relevant concentration of an ImmTAV molecule. We have included the data below and in a new supplementary table 2.

pHLA-E*01:03	pHLA loaded (RU)	a09b08 binding (RU)		
		Run 1	Run 2	Run 3
Leaders A2/Cw3/G	2090	-3.3	-3.2	-3
Leaders A34/A80/B7/Cw7	2097	-4.8	-4.7	-4.5
Env ₃₇₁₋₃₇₉ (L6I)	687	103.4	138.9	162.8

For the functional assays, it is difficult to assess how the amount of output in the ELISPOT or cytokine release assays compares to (for example) targeting an HLA-A2-restricted response. A direct comparison with a previously described ImmTAV would be helpful calibration.

The viral inhibition results in Figure 6G-H show what appear to be an extremely modest effect. Can the authors provide some type of comparison with another ImmTAV? And could a negative control be included? I would also consider moderating language about these results given the largest outcomes seem like ~2-4 fold differences.

We agree that the viral inhibition was modest and believe that this could reflect the short co-culture (3 days). We have therefore repeated the viral inhibition assay with a longer time course. Wildtype (natural HLA-E expression), IFN γ treated or HLA-E*01:03 transfected HepG2 cells were used as targets, and we included two additional controls: an irrelevant HLA-E ImmTAB molecule (Paterson et al., PMID: 38691588) and an HBV Env-specific HLA-A*02:01-

restricted ImmTAV molecule previously described in Fergusson et al. (PMID: 32770836). As shown in the new figure 6, we observed a greater reduction of HBV replication after 4 and 6 days of co-culture (3 to 4-fold reduction in co-cultures with HepG2 pre-treated with IFN- γ at day 4 which increases to 10-fold at day 6; 10-fold reduction in co-cultures with HLA-E*01:03 HepG2 at day 4 which increases to 15-fold at day 6). Additionally, our HLA-E a09b08 ImmTAV molecule induced similar reduction of viral replication to the HBV HLA-A*02:01 ImmTAV molecule when tested against targets overexpressing HLA-E*01:03. However, only the HBV HLA-A*02:01 ImmTAV molecule reduced viral replication in response to WT HepG2, likely due to lower HLA-E-HBV-Env peptide density on the cell surface. The irrelevant HLA-E ImmTAB molecules showed no reduction of viral replication to any of the HepG2 cells targets. Similar results were observed for IFN- γ and granzyme B release. Below is a copy of the new Figure 6 for reference.

Fig. 6: ImmTAV activates T cells to eliminate HBV-transfected HepG2 cells. **a** Surface expression levels of HLA-E on HepG2 cells were analysed by flow cytometry. HepG2 wildtype and HLA-E*01:03 overexpressing cells were transfected with a replication competent HBV plasmid encoding genotype A2 (Env371-379 (L6I)). **b** 24 h post-transfection, some wells of HepG2 wildtype cells were stimulated with IFN- γ (1 ng/mL) for 16 h. Cells were washed and co-cultured with pan T cells from three donors (1:1 ratio) with or without a09b08 ImmTAV, HBV HLA-A*02:01 ImmTAV or Mtb RLPA HLA-E ImmTAB (negative control) at 1 nM and 10 nM. Culture supernatants were harvested day 4 and day 6 according to the schematics shown. Levels of IFN- γ (**c**) and granzyme B (**d**) in the culture supernatants at day 4 were quantified using MSD. Data represents the mean \pm SEM of triplicates (n=3). HBeAg (**e, f**) and HBsAg (**g, h**) levels in the culture supernatant at day 4 and day 6 were quantified using ELISA. Data are represented as mean \pm SEM of duplicates from 3 donors. Significant differences are * p < 0.05, ** p < 0.001, **** p < 0.0001 by ANOVA followed by Tukey's post-hoc test.

Given that HBV Env371-379 L6I was the only variant tested which resulted in T cell activation, cytokine secretion and antiviral activity in the viral infection model (albeit in the presence of IFN- γ pre-treatment), could the authors elaborate further on the prevalence of this particular variant in the discussion? For example, what would the associated therapeutic coverage among individuals infected with HBV look like?

We have determined the HBV peptide variant prevalence across the five major HBV genotypes (A-E) and the estimated coverage of people with CHB of Env₃₇₁₋₃₇₉ (L6I) variant worldwide. The data are now shown in a new supplementary table 1 and data 2, and the methods included in the supplementary file.

Line 154: *“To understand the potential therapeutic relevance of our findings, we determined the genotype prevalence of the HBV Env371-379 peptides across the five genotypes A-E. We calculated that the Env₃₇₁₋₃₇₉ peptide is the most prevalent in genotypes C and D (65% and 84% of the sequences analysed, respectively), while the Env₃₇₁₋₃₇₉ (S3N) is present in genotype C (21%) and the Env₃₇₁₋₃₇₉ (L6I) is most prevalent in genotypes A and E (45% and 91% respectively) (Supplementary Table 1).”*

Line 448: *“Because of the distinct geographical distribution of the different HBV genotypes, we estimate that percentage of CHB patients’ coverage of the most stable Env371-379 (L6I) variant worldwide would be around 29% (Supplementary Data 2).”*

HLA-E/HLA-B*07:01 specific CD8⁺ T cells have been reported to be present at high frequencies, averaging 10% of total CD8⁺ T cells in CMV⁺ donors (Sullivan et al Sci Immunology 2021). Therefore, it would be nice if the authors included HLA-E/HLA-B*07:01 dextramers as negative controls in their in vitro priming and subsequent staining experiments to rule out cross-reactivity. Dextramer staining to assess potential TCR cross-reactivity between HLA-E vs HLA-A2 presenting the HBV Env variants would also constitute important controls here as this peptide binds to HLA-A2 and could explain why HBV⁺ donors had higher dextramer staining relative to healthy donors if prior expansion was in fact being driven by HLA-A2.

This is an excellent suggestion; however, we were not able to repeat the in vitro priming experiment due to a lack of additional PBMC samples from the CHB patients. Instead, we now show as additional panel in supplementary Figure 13b that HLA-E/Cw3 and HLA-A2 (L6I) dextramers do not stain Jurkat cells transduced with the WT or affinity enhanced TCR. Of note, only two of the patients with chronic HBV infection were HLA-A*02:01, 98188 and 10727 and in neither we could detect L6I as viral variant (Supplementary Table 8).

b Jurkat clones transduced with wild type (WT) and high affinity a09b08 TCR (L6I) were stained with dextramer-HLA-E Env371-379 (L6I) (pink and purple histograms), HLA-A*02:01 Env371-379 (L6I) (yellow and dark green histograms), or the HLA-E*01:03 Cw3 signal peptide (black and brown histograms) to test the dextramer specificity.

In the discussion the authors mention that mass spec analyses were conducted to identify HLA-E presentation of HBV Env peptide variants but that even the highest affinity L6I variant was almost undetectable via this approach. Would the authors consider including these data in the supplementary? For example, it would be helpful to the field to know whether the authors used peptide pulsing versus endogenous peptide presentation methods and whether highly sensitive SIL approaches were attempted?

We thank the reviewer for their interest in our mass spec analysis which is now included in the revised manuscript. For identification and quantification of HBV Env₃₇₁₋₃₇₉ (L6I), we used a targeted parallel reaction monitoring (PRM) mass spectrometry method and a stable heavy isotope labelled internal peptide standard. We analysed PLC/PRF/5 ($4-5 \times 10^8$) cells either wild type (WT) or transduced with HLA-E*01:03. Peptide HLA complexes were enriched and the HBV L6I variant levels assessed by PRM. No L6I was detected in WT PLC/PRF/5 cells. A few fragment ions corresponding to HBV L6I was identified when HLA-E levels were increased by HLA-E*01:03 overexpression in PLC/PRF/5 cells. The inability to detect HBV L6I by mass spectrometry in WT PLC/PRF/5 cells could, in addition to low abundance, be due to losses during the affinity enrichment procedure, which is overcome by HLA-E*01:03 overexpression.

HBV Env₃₇₁₋₃₇₉ (L6I) peptide identification by mass spectrometry. Fragment ion intensities obtained when analysing (a) wild type PLC/PRF/5 cells or (b) PLC/PRF/5 cells overexpressing HLA-E*01:03 by parallel reaction monitoring mass spectrometry. (c) Absolute amount of peptide detected in samples when using a ratio metric approach comparing endogenous peptide levels to the stable heavy isotope labelled internal peptide standard.

In the discussion, the section: ‘Taken together, our findings demonstrate that the HBV Env371-379 (L6I) peptide can compete with VL9 variants to bind HLA-E and be targeted with TCR-based immunotherapeutics.’ should be softened or followed with the caveat that such immunotherapeutic targeting would likely only be effective in combination with IFN- γ treatment or in a setting where HLA-E surface expression on HBV-infected cells was elevated.

We thank the reviewer for this suggestion. We have modified the discussion section, which now reads “...and be targeted with TCR-based immunotherapeutics when increased surface levels of HLA-E are present in cells”.

Minor points

The FACS histograms in Supplementary Figure 1 which depict fluorescence intensity are more informative in terms of HLA-E expression density than the plots in Figure 1B which simply portray the % of cells in the HLA-E+ gate. The authors might want to consider switching these plots between the main and supplementary figure packets.

We have modified Fig. 1b as suggested by the reviewer, to show the FACS histograms plots of one dataset. The histogram plots with Geometric mean fluorescent values of triplicate samples of two independent experiments are shown in the new Supplementary Figure 1.

13. The wording of the sentence between lines 174-177 is slightly unclear.: 'The peptide contacts are predominantly through hydrophobic interactions, including the TCR alpha chain H94 stacking with I1 and P4...'. It sounds as though the authors are stating a list of hydrophobic interactions between the TCR and peptide. However, the final contact mentioned in this list is between the TCR beta chain Arg 95 and peptide Leu 8 which appears to be a hydrogen bond, not a hydrophobic interaction?

The contact between Arg95 and Leu8 is not a H-bond as the closest contacts are between two carbon atoms (CD atom in Arg and CD1 atom in Leu), so this was mingled with other van der Waals contacts. The position of atoms varies in the L6I variant, but it is still not a H-bond.

Reviewer 3

Major questions

A. Regarding the presentation of the Env371-379/HLA-E target and relative stabilities:

From the perspective of stability, the authors showed in both Table S1 and Figure 1A that the thermal stability of the original Env371-379 peptide in complex with HLA-E was around 47 °C. A measurement by SPR showed a $t_{1/2}$ of 6.7 min for this peptide. How does this compare with other naturally presented HLA-E peptides, such as the signal peptide? Although the SPR measurement showed the S3N and especially the L6I variants have longer half-lives than the original Env371-379, are there T_m values for them as well – if not, these should be included? For the SPR stability measurement using ILT2 protein, is there any control measurement testing the stability of either Mtb RLPA or signal peptides in complex with HLA-E as a reference? This is important for the points below.

As suggested by the reviewer, in the revised manuscript we have modified Figure 1c to include the T_m and $t_{1/2}$ values of HLA-E*01:03 in complex with two leader peptides (Cw3 and A1), in addition to the three refolded HLA-E*01:03 HBV Env₃₇₁₋₃₇₉ peptide variant complexes. We demonstrate a comparable $t_{1/2}$ of 2 hours at 25°C for L6I and leader peptides, and all three complexes have similar T_m values of 49°C.

Differential scanning fluorimetry (DSF) of refolded HLA*01:03 HBV Env₃₇₁₋₃₇₉ and Env₃₇₁₋₃₇₉ (L6I) complexes. Melting curves (top panel) and melting peaks (bottom panel), pHLA-E complexes melting temperature (T_m) are shown in the top panel.

Following these questions, in presentation by HLA-E, the selected HBV peptide needs to compete with the existing signal peptides in HLA-E presentation. The relative binding values are important here, as is the comment in the discussion about detection via mass spectrometry. Have the authors pulsed the HLA-E-positive cells with a mixture of HBV and VL9 peptides at different ratios in the function analyses? Overall, the authors need to perform a stronger job at ascertaining the target and potential competitors/cross-reactive epitopes (see cross-reactivity point below).

We appreciate the reviewer's suggestion to pulse HLA-E-positive cells with a mixture of HBV and VL9 peptides at different ratios in the functional assays. However, we have demonstrated (Fig. 5a) that the HBV Env₃₇₁₋₃₇₉ L6I peptide, which displays equivalent stability to leader peptides (Fig 1c), is capable of competing with leaders and can be targeted by our ImmTAV molecule under physiological conditions (lysis of PLC/PRF/5 target cells). We have also demonstrated (Fig. 4c) that, when incubated with HepG2 target cells endogenously expressing the HBV Env₃₇₁₋₃₇₉ L6I peptide, the a09b08 ImmTAV molecule induces specific IFN- γ release from healthy donor PBMC. We believe these results clearly demonstrate the ability of HBV Env₃₇₁₋₃₇₉ L6I peptides to compete with endogenous VL9 peptides for HLA-E presentation.

The authors also mention that the Env₃₇₁₋₃₇₉ peptide can be presented by HLA-A*02:01 allele as well. What is the stability of the various peptides in complex with HLA-A*02:01? Is there a possibility that the Env₃₇₁₋₃₇₉ L6I peptide is preferentially presented by HLA-A*02:01, and could this be a limiting factor in targeting?

We understand and share the reviewers concerns regarding the fact that HBV Env₃₇₁₋₃₇₉ peptide can also be presented by HLA-A*02:01 and if this could be a limiting factor in targeting this HLA-E epitope.

As suggested by the reviewer we have determined the stability of the HBV Env₃₇₁₋₃₇₉ peptides in complex with HLA-A*02:01. In the revised manuscripts we have modified supplementary

figure 3a (see below) to include the T_m and $t_{1/2}$ values of HLA-A*02:01 in complex with the three HBV Env₃₇₁₋₃₇₉ peptide variants. The overall stability of these molecules ranges around 17-19h when measured at 25°C, with a T_m of 63°.

a

pHLA-A*02:01	pHLA loaded (RU)	Folded pHLA (%)	a09b08 binding (RU)	pHLA ($t_{1/2}$ h)	pHLA (T_m °C)
Env ₃₇₁₋₃₇₉	95	30.7	0	17.2±7.7	63.9±0.1
Env ₃₇₁₋₃₇₉ (S3N)	110	26.5	0.5	16.8±7.5	63.6±0.1
Env ₃₇₁₋₃₇₉ (L6I)	98	29.8	1.1	19.0±7.4	65.1±0.1

We discuss the possibility of HLA-A*02:01 presentation of the Env₃₇₁₋₃₇₉ peptide and provide experimental evidence that the a09b08 ImmTAV does not bind HBV Env₃₇₁₋₃₇₉ peptides in complex with HLA-A*02:01 (Supplementary Figure 3). Accordingly, ImmTAV-dependent IFN- γ release against HLA-A*02:01 positive HepG2 targets transfected with minigenes was not inhibited by the anti-HLA-A2 antibody (Figure 4d). It is possible that Env₃₇₁₋₃₇₉ L6I peptide binding to HLA-A*02:01 might be a limiting factor in targeting with the a09b08 ImmTAV if HLA-E expression is low. Indeed, we show that when HepG2 cells are transfected with replication-competent infectious 1.3-mer cDNA clones, Env₃₇₁₋₃₇₉ L6I is efficiently presented only when cells are pretreated with INF- γ to increase HLA-E expression or are transduced with lentiviral particles encoding HLA-E. Additionally, presentation of Env₃₇₁₋₃₇₉ or Env₃₇₁₋₃₇₉ S3N is not detected in these experimental models, and we discuss the possible explanations, namely, the low affinity of these peptides for HLA-E molecules, the inability to compete with VL9 peptides and/or the potential sequestration by HLA-A2 molecules.

B. Regarding cross-reactivity

Although the authors have put in considerable efforts for minimizing the potential cross-reactivity of a09b08TCR/ImmTAV towards other HLA-E presented peptides, is there a possibility that the a09b08TCR can cross-react with other peptide/HLA-E antigens with an affinity weaker than that for the Env peptides/HLA-E but still above the threshold to activate T cells, especially with high levels of antigen presentation? For example, in Table S5, although the INF-g release for other peptides are quite low compared to the Env₃₇₁₋₃₇₉ (L6I), there is still some signal with 1 nM usage of ImmTAV.

We understand the reviewer concerns about a09b08 TCR cross-reactivity with other peptide/HLA-E complexes. We have tested the binding affinity of the a09b08 ImmTAV, at 8.5 nM concentration, against a panel of pHLA-E/VL9 complexes and no binding was observed. We now include a new supplementary table 2 with these data.

The ELISpot responses shown in the old Table S5 (now Supplementary Table 7) were performed in non-physiological conditions (HLA-E overexpression, excess peptide (10 μ M) and high ImmTAV (1nM)). Nevertheless, a ~30-fold window was observed. In all other experiments, negligible background was observed in the absence of the cognate antigen Env₃₇₁₋₃₇₉. We believe that all together these results demonstrate lack of cross reactivity of the a09b08 ImmTAV.

This is particularly relevant given 1) the questions regarding HLA-E/peptide stability

above, 2) the statement on lines 168-171 that “the three HBV Env peptides adopt a similar binding conformation to the previously described binding of Mtb44 and canonical VL9 signal peptides to HLA-E”, and 3) the TCR’s picomolar affinity to the HLA-E/Env peptides. If there is really no detectable cross-reactivity to those epitopes, the authors should address why or present their conclusions with more caution. Clearly, the potential for cross-reactivity is crucial for any therapeutic development and this should be addressed very carefully.

We understand the reviewer’s concerns for the potential cross-reactivity which could hinder any therapeutic development. We have demonstrated systematically that the a09b08 ImmTAV does not bind HLA-E*03:01/VL9 or HLA-A*02:01/L6I complexes either by SPR or Jurkat display system and does not induce appreciable T cell activation. These results are in agreement with the reported specificity of the ImmTAB molecule against the HLA-E/Mtb44 complex, a peptide which also shares with VL9 the binding conformation to HLA-E (PMID: 38691588). These results and the recent FDA approval of Tebentafusp, the first soluble TCR bispecific therapeutic, indicate that it is possible with our platform to develop high affinity TCR molecules capable of distinguishing cognate pHLA complexes from peptides with similar HLA binding conformations. Nevertheless, we agree with the reviewer that further systematic screening with large panels of normal cell lots would need to be conducted prior to consideration for a first-in-human clinical trial.

C. On the structures:

In Table S2, the l/σ for the outer shell is different from those values in the validation reports and are all quite low (~ 0.5). Can the authors double check those numbers? Also, if the numbers in Table S2 are correct, does the highest resolution shell with l/σ around 0.5 really help with the improvement of structure refinement, or is this essentially noise? The value of 0.5 is well below what is normally used as a cutoff.

We used the CC1/2 cutoff of 0.3+ for the highest resolution shell. This metric has been accepted in recent years. The reported $1/\sigma$ values now in supplementary table 3 came from xia2 dials automated processing, and in the validation reports they were calculated using Xtriage by the PDB deposition server. It is not clear why there are significant differences in values between the two different programs. When we tried to lower resolution during refinements, the RMS deviations (bonds and angles) generally became worse.

In lines 166-168, the authors gave some explanation for why the S3N variant is more stably presented by HLA-E than the Env371-379. However, according to the stability measurement by SPR, the L6I stability ($t_{1/2} \sim 2$ h) increased the most from the Env371-379 ($t_{1/2} \sim 6.7$ min) comparing to the S3N variant ($t_{1/2} \sim 39.5$ min). Is there any structural reasoning for the higher stability of L6I?

This comment was also raised by reviewer 2. As mentioned in the previous section, we have now included a supplementary figure 5 comparing S3N and L6I variants with the index peptide surrounding peptide positions 3 and 6, respectively. The subtle differences surrounding peptide position 6 and HLA residues T70 and F74 might potentially explain the higher stability for the L6I variant.

For the statement on lines 168-171, can the authors support it with an alignment of the Env peptides/HLA-E and Mtb44 and canonical VL9 signal peptides/HLA-E structures?

We have added a supplementary figure 6 to compare the Env₃₇₁₋₃₇₉ peptide with Mtb44 and VL9 signal peptide.

-In Figure 3A, the authors should show the peptide density map to support the peptide structural comparison. This should be an omit map to avoid bias, and at a size to clearly assess density and fit.

For clarity, we have included a supplementary figure 4 for the omit maps of peptide variants.

-In Figure 3C, some amino acid labels are incorrectly indicated: Env371-379 structure: L5; Env371-379 L6I structure: L6.

We thank the reviewer for noticing this. We have generated a new figure 3 to correct this and to accommodate other fig. 3 comments.

D. Other receptor reactivity:

Since HLA-E is a ligand for NK cells, does the ImmTAV influence NK cell function via any NK receptor (CD94/NKG2A, B, C, etc.) interactions with HLA-E?

We observed no binding of the soluble forms of the CD94/NKG2A and -NKG2C NK cell receptors ectodomains to the HBV Env₃₇₁₋₃₇₉ peptides in complex with HLA-E*01:03, suggesting that the ImmTAV is unlikely to have a direct impact on NK cells functions mediated by these two receptors. These results agree with what was previously reported by Paterson et al. (PMID: 38691588) with an ImmTAB against the Mtb RLPA peptide in complex with HLA-E*01:03.

We have not included these results in the revised version of the manuscript but could add them if this reviewer felt it was needed.

pHLA-E*01:03 complexes

ILT2 injection at end of run:

Binding of soluble forms of the CD94/NKG2A and CD94/NKG2C NK cell receptors ectodomains to HLA-E*01:03 HBV Env₃₇₁₋₃₇₉ peptide complexes or HLA-E*01:03 control complexes. Top row: binding of CD94/NKG2A and -NKG2C molecules to control HLA E*01:03 complexes (left to right: blank flow cell, A1 leader peptide, Cw3 leader peptide, and Mtb44 RLPA peptide). Middle row: binding of CD94/NKG2A and -NKG2C molecules to HLA E*01:03 HBV Env₃₇₁₋₃₇₉ peptide complexes (left to right: Env₃₇₁₋₃₇₉, Env₃₇₁₋₃₇₉ (L6I), and Env₃₇₁₋₃₇₉ (S3N)). Bottom row: ILT2 injection at the end of the run (left to right: blank flow cell, Cw3 leader peptide, A1 leader peptide, Mtb44 RLPA peptide, HBV Env₃₇₁₋₃₇₉, HBV Env₃₇₁₋₃₇₉ (L6I), and HBV Env₃₇₁₋₃₇₉ (S3N)).

Minor comments:

Line 81: The usage of term “HBeAg” is confusing without any explanations for what it stands for.

We have modified the revised manuscript in lines 83-84 to clarify the term HBeAg.

Lines 117-119: the statement, “this peptide is presented in all forms of envelope proteins (large, medium, and small)” is misleading (overemphasized) without mentioning its prevalence in different genotypes. Can the authors provide accession numbers for some of the sequences used as an example to support the statement?

The peptide Env₃₇₁₋₃₇₉ itself is present in all forms of envelope proteins (large, middle, and small), however the variants appear with different frequency in the HBV genotypes. We added to the revised manuscript the estimated genotype prevalence of the three HBV peptides (Line 156). The method describing how the genotype prevalence was calculated was added to the supplementary file. The accession numbers used are described in Bell et al. 2016. We have also added the peptides AA numbers to the supplementary data 1. Peptide amino acid numbering was retrieved from a full-length HBV surface antigen protein (Uniprot: P17398).

Line 162: The authors should indicate somewhere in the main text about which HLA-E allele was solved in crystal structures. The only position I could clearly find an answer to this is in the PDB validation report.

We have modified the manuscript to indicate that the HLA-E allele 01:03 was solved in crystal structures. Line 175 of the modified manuscript now reads “The crystal structures of a09b08 TCR in complex with all three HBV pHLA-E*01:03 complexes...”

Lines 179-180: I think the statement from the authors that “the interactions between the a09b08 TCR and the HLA-E heavy chain are the same with all three peptides” is a little casual, as Fig.3D indicates residues of the TCR only interacting with one of the three peptide/HLA-E targets. Even though the contribution of these interactions can be limited, they’re still considered to be different unless the authors can provide an explanation for why they’re not significant.

We thank the reviewer for noticing this. We have now removed this statement and instead included a supplementary table 4 which lists all the interactions to highlight the minor differences observed between TCR and HLA-E in the three complexes based on distance cut-off.

Figure 2C: The unit for standard deviations looks to be M in this format.

We thank the reviewer for their careful examination of the figure, we have modified Fig. 2c in the revised manuscript.

Figure 3A: It’s quite untraditional to see pMHC in a top-down view as shown. I understand that the authors may have their own considerations on this strategy, but this is difficult to interpret. If the point is to show the similarity of the three peptide structures, authors should show a structural alignment of all peptides without the distraction from the HLA-E binding groove.

We have modified the alignment of the three HBV peptides in Fig. 3 to remove the HLA groove.

Figure S2A: It could be due to the low resolution of figure, but I could not see error bars in all the figures.

We thank the reviewer for their careful examination of the figures, a higher resolution figure for publication has been provided and should resolve any apprehensions over figure resolution. The error bars have been added to all the figures, however, in some graphs, the error bars are too small to be seen.